# Economic Potential of Bio-Ethylene Production via Oxidative Coupling of Methane in Biogas from Anaerobic Digestion of Industrial Effluents

**Alberto Teixeira Penteado [1,\*]** , **Giovanna Lovato [2,3]** , **Abigail Pérez Ortiz [1]** , **Erik Esche [1]** , **José Alberto Domingues Rodrigues [2]** , **Hamid Reza Godini [1,4]** , **Alvaro Orjuela [5]** , **Jūlija Gušča [6]** and **Jens-Uwe Repke [1]**

1   Group of Process Dynamics and Operations, Technische Universität Berlin, Sekretariat KWT9, Straße des 17 Juni 135, 10623 Berlin, Germany; a.perezortiz@tu-berlin.de (A.P.O.); erik.esche@tu-berlin.de (E.E.); h.r.godini@tue.nl (H.R.G.); jens-uwe.repke@tu-berlin.de (J.-U.R.)
2   Biochemical Engineering Laboratory, Mauá School of Engineering, Mauá Institute of Technology, Praça Mauá 1, São Caetano do Sul 09580-900, SP, Brazil; glovato@sc.usp.br (G.L.); rodrigues@maua.br (J.A.D.R.)
3   São Carlos School of Engineering, University of São Paulo, Avenida Trabalhador São-Carlense, São Carlos 13566-590, SP, Brazil
4   Inorganic Membranes and Membrane Reactors, Department of Chemical Engineering and Chemistry, Eindhoven University of Technology (TU/e), Den Dolech 2, 5612 AZ Eindhoven, The Netherlands
5   Department of Chemical and Environmental Engineering, Universidad Nacional de Colombia, Bogotá 111321, Colombia; aorjuelal@unal.edu.co
6   Institute of Energy Systems and Environment, Riga Technical University, 12-K1 Āzene Street, LV-1048 Riga, Latvia; julija.gusca@rtu.lv
\*   Correspondence: alberto.penteado@tu-berlin.de or albertopenteado@gmail.com

**Abstract:** Brazil's large biofuels industry generates significant amounts of effluents, e.g., vinasse from bioethanol, that can effectively be used as substrate for production of biogas via Anaerobic Digestion (AD). The Oxidative Coupling of Methane (OCM) is the heterogeneous catalytic oxidation of methane into ethylene, which is a main building block for the chemical industry. This work investigates the potential and competitiveness of bio-ethylene production via OCM using biogas produced by biological anaerobiosis of vinasse as a feedstock. The proposed process can add incentive to treat of vinasse via AD and replace fossil ethylene, thus potentially reducing emissions of Greenhouse Gases (GHG). A process model is developed in Aspen Plus v10 software and used to design an economic Biogas-based Oxidative Coupling of Methane (Bio-OCM) process that consumes biogas and oxygen as educts and produces ethylene, ethane, and light off-gases as products. Operating conditions in the reaction section are optimized and a reaction product yield of 16.12% is reached by applying two adiabatic Packed Bed Reactors (PBRs) in series. For the downstream $CO_2$ removal section, a standalone amine-absorption process is simulated and compared to a hybrid membrane-absorption process on an economic basis. For the distillation section, two different configurations with and without Recycle Split Vapor (RSV) are simulated and compared. The bio-ethylene production cost for a Bio-OCM plant to be installed in Brazil is estimated considering a wide range of prices for educts, utility, side products, and equipment within a Monte Carlo simulation. The resulting average production cost of bio-ethylene is $0.53 \pm 0.73 \, \mathrm{USD \, kg_{C_2H_4}^{-1}}$. The production cost is highly sensitive to the sales price assigned to a light off-gas side-product stream containing mostly the un-reacted methane. A sales price close to that of Brazilian pipeline natural gas has been assumed based on the characteristics of this stream. The Monte Carlo simulation shows that a bio-ethylene production cost below or equal to $0.70 \, \mathrm{USD \, kg_{C_2H_4}^{-1}}$ is achieved with a 55.2% confidence, whereas market values for fossil ethylene typically lie between $0.70 \, \mathrm{USD \, kg_{C_2H_4}^{-1}}$ to $1.50 \, \mathrm{USD \, kg_{C_2H_4}^{-1}}$. Technical and economic challenges for the industrial implementation of the proposed Bio-OCM process are identified and relevant opportunities for further research and improvement are discussed.

**Keywords:** anaerobic digestion; biogas application; ethylene; methane; oxidative coupling of methane

## 1. Introduction

The Anaerobic Digestion (AD) of biodegradable wastes is a bio-process in which the organic matter in effluents and residues is ultimately converted to methane ($CH_4$) and carbon dioxide ($CO_2$). Biogas from AD typically contains 50% to 70% $CH_4$, 30% to 50% $CO_2$, $\leq$1% $N_2$, and 10 ppm$_v$ to 2000 ppm$_v$ $H_2S$ depending on the substrate and processing conditions [1–3].

AD is commonly described by four major steps performed by microorganisms and driven by thermodynamic principles: (i) hydrolysis, (ii) acidogenesis, (iii) acetogenesis and (iv) methanogenesis. Since microorganisms are not able to assimilate particulate organic matter, the first stage in AD is the hydrolysis of complex particulate materials into simpler dissolved materials, which can cross the cell walls of fermentative bacteria. This conversion of particulate materials into dissolved materials is achieved through the action of exoenzymes excreted by hydrolytic bacteria. Soluble products from the hydrolysis phase are metabolized inside the cells of fermentative bacteria, being converted into several simpler compounds, which are then excreted by the cells. The compounds produced include: volatile fatty acids, alcohols, lactic acid, carbon dioxide, hydrogen ($H_2$), ammonia ($NH_3$) and hydrogen sulfide ($H_2S$), in addition to new bacterial cells. As volatile fatty acids are the main product of fermentative organisms, they are usually called acidogenic fermentative bacteria. Acetogenic bacteria are responsible for oxidizing the products generated in the acidogenic phase into suitable substrates for methanogenic microorganisms. The products generated in this process are $H_2$, $CO_2$, and acetate. The final step in the global process of anaerobic degradation, methanogenesis, is the conversion of acetate into $CH_4$ and $CO_2$ by aceticlastic methanogens and of $H_2$ and $CO_2$ into $CH_4$ by hydrogenotrophic methanogens.

Biogas can be used for a number of purposes, including electricity production (most common), heat generation and as raw material for industries [1–3]. Biogas production has been a reality since the 1930s for the stabilization of sewage sludge. What has changed over the years is that its production has been optimized and achieved at industrial scale with a higher efficiency, degree of complexity and specification, particularly in developed countries. Biogas production can reduce emission of Greenhouse Gases (GHG), provide a renewable source of energy and reduce impacts of pollution by waste disposal [4]. Furthermore, the activation of methane and/or carbon dioxide present in biogas enables its chemical conversion into higher value biochemicals and biofuels [5].

One possible process for methane conversion into a valuable chemical is the Oxidative Coupling of Methane (OCM), which is its heterogeneous catalytic oxidation into ethylene: a major feedstock for chemical and polymer production. It is of scientific consensus that OCM occurs via three steps (simplified): (i) activation of methane to methyl radical through a C–H bond breaking and a hydrogen abstraction, (ii) homogeneous coupling of two methyl radicals to ethane in the gas phase, and (iii) oxidative dehydrogenation of ethane to ethylene [6]. These steps are summarized in Equations (2) and (5). However, many parallel side reactions also occur, which adds complexity to the downstream separations. The full reaction network considered in this study, as proposed by [7], is given in Equations (1)–(10). A trade-off between methane conversion and selectivity towards $C_2$ products (ethane and ethylene) is observed, which is typical of selective oxidation reactions.

$$CH_4 + 2\,O_2 \longrightarrow CO_2 + 2\,H_2O \tag{1}$$

$$2\,CH_4 + \frac{1}{2}\,O_2 \longrightarrow C_2H_6 + H_2O \tag{2}$$

$$CH_4 + O_2 \longrightarrow CO + H_2O + H_2 \tag{3}$$

$$CO + \frac{1}{2}\,O_2 \longrightarrow CO_2 \tag{4}$$

$$C_2H_6 + \frac{1}{2}\,O_2 \longrightarrow C_2H_4 + 2\,H_2O \tag{5}$$

$$C_2H_4 + 2\,O_2 \longrightarrow 2\,CO + 2\,H_2O \tag{6}$$

$$C_2H_6 \longrightarrow C_2H_4 + H_2 \tag{7}$$

$$C_2H_4 + 2\,H_2O \longrightarrow 2\,CO + 4\,H_2 \tag{8}$$

$$CO + H_2O \longrightarrow CO_2 + H_2 \tag{9}$$

$$CO_2 + H_2 \longrightarrow CO + H_2O \tag{10}$$

OCM is not yet commercially applied, but the surge in shale gas exploration has made it a potential route for producing valuable ethylene from cheap methane sources and without wildly fluctuating prices of crude oil [6]. A demonstration plant has been built and put into operation by Siluria Technologies in Texas, U.S. [8].

In order for the ethylene production from biogas via OCM to be economically competitive with the consolidated production from oil [9], a significant amount of biogas (process feed) is required to merely approach the economies of scale typical of this industry. Consequentially, a great amount of a biodegradable effluent (substrate) must be available for AD. Agroindustry wastes such as whey (dairy industry effluent), glycerin (biodiesel production effluent), and sugarcane stillage (also called vinasse, which is the sugar and ethanol production effluent) are all produced in large quantities worldwide and, due to their high organic load, pose an environmental threat if not properly treated. Several studies have shown the feasibility of treating these effluents by anaerobiosis with the associated production of bio-energy, i.e., methane and/or hydrogen [10–17].

Among these effluents, vinasse stands out as a particularly good candidate for large-scale biogas generation. It is the main liquid stream from first-generation ethanol production process from sugarcane, beet, sweet sorghum, grape, corn, wheat, rice, cassava, potato, and others [18]. Together, the U.S. and Brazil produced 85% of the world's ethanol in 2017. The vast majority of U.S. ethanol is produced from corn, while Brazil primarily uses sugarcane [19].

Considering the Brazilian scenario, vinasse is derived from the ethanol distillation step, leaving the columns at $\approx$360 K. The presence of melanoidins and the high organic acid content gives it a dark-brownish color and low pH, respectively. Sugarcane processing plants usually generate from 10 L to 15 L of vinasse per L of produced ethanol and, in 2019, the Brazilian Ministry of Agriculture, Livestock and Food Supply estimated a production of $31.6 \times 10^9$ L of ethanol [20,21]. This leads to an estimated $31.6 \times 10^{10}$ L to $47.4 \times 10^{10}$ L of vinasse in this country alone in a single year. The vinasse is currently applied for fertilization, but if not properly conditioned, vinasse may lead to pollution of soil and water bodies. AD is an effective way to reduce its high organic load, i.e., COD, which may reach values up to $65{,}000\,\text{mg}_{O_2}\,\text{L}^{-1}$, and make it suitable for use in fertilization while simultaneously producing biogas [22].

The Southeast region of Brazil and, notably the state of São Paulo, concentrates a great portion of the bioethanol production in the country. An interactive map has recently been created by researchers at University of São Paulo containing the availability of organic wastes (substrates), the biogas and biomethane production potential, as well as the available infrastructure such as pipelines and compression stations [23]. The total estimated biogas production potential lies at around $16.8 \times 10^9\,\text{Nm}^3\,\text{year}^{-1}$ ($8.9 \times 10^9\,\text{Nm}^3_{CH_4}\,\text{year}^{-1}$) with

around 88 % of the total stemming from vinasse AD [23]. Based on the yields obtained in the present study ($36.25 \, \mathrm{kg_{C_2H_4} \, kg_{CH_4}^{-1}}$), this could be used to produce around 900 kt of ethylene per year and potentially replace some $9 \, \mathrm{Mt \, year^{-1}}$ of oil (assuming $0.28 \, \mathrm{t_{C_2H_4} \, t_{Naphtha}^{-1}}$ and $0.35 \, \mathrm{t_{Naphtha} \, t_{Oil}^{-1}}$). This region is also very industrialized and hosts 42% of the total Brazilian population [24], therefore also containing the infrastructure and market demand that are required for a project such as this.

　　　This contribution investigates the potential and competitiveness of industrial bio-ethylene production via OCM using biogas produced by biological anaerobiosis of sugarcane vinasse as a feedstock. The main goals are to apply process simulation models to conceptually design an economic Biogas-based Oxidative Coupling of Methane (Bio-OCM) process using the structure depicted in Figure 1 and to assess its techno-economic feasibility. A model of the OCM reaction section applying adiabatic Packed Bed Reactors (PBRs) is used to maximize $C_2$ product yield by manipulating the operating conditions. For the $CO_2$ removal section, a superstructure optimization is applied to determine whether to use a standalone amine-absorption configuration or a hybrid configuration employing both membrane and absorption. Two different process configurations are compared for the distillation section. The first distillation configuration only applies external low-temperature refrigeration, while the second distillation configuration adiabatically expands fractions of the process streams to reduce external refrigeration consumption. Once the optimal process design is defined, the bio-ethylene production cost is estimated and compared to typical market values for fossil ethylene. Finally, a Monte Carlo simulation is performed to encompass uncertainties in the cost estimations.

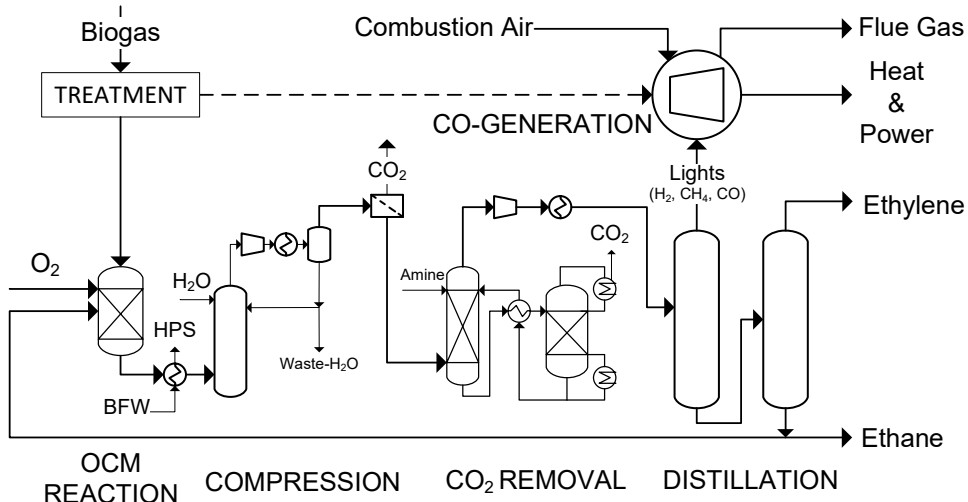

**Figure 1.** Process flow diagram of the Bio-OCM process. Dashed lines represent optional streams and units. Reproduced with permission from Penteado, Ph.D. Thesis, TUB, 2021 [25].

## 2. Materials and Methods

　　　The process flow diagram for the Bio-OCM process considered herein is depicted in Figure 1. The process plant consumes biogas and oxygen as educts, while producing ethylene as the main product and ethane and light off-gases as side-products. Biogas is first subjected to a regular treatment step for the removal of impurities such as $H_2S$ and $NH_3$. The $CO_2$ removal step, i.e., biogas upgrade, is avoided and the $CO_2$ present in biogas serves to dilute the educts in the feed stream to the OCM reactor. This helps to contain the intense reactions' heat release. The treated biogas stream is fed to the OCM reactor together with an oxygen stream. The hot reaction gases are cooled by generating High Pressure Steam (HPS) and compressed prior to the $CO_2$ removal section. $CO_2$ removal is achieved either by a hybrid permeation-absorption process or by a standalone amine-absorption process. The final hydrocarbon separation is achieved by distillation. The off-gas (lights)

stream containing mostly the unreacted methane is exploited energetically in a Combined Heat and Power (CHP) unit. The process model focuses on the three main sections of the proposed Bio-OCM process, i.e., reaction section, compression and $CO_2$ removal section, and distillation section, while the vinasse treatment by AD, the CHP, and the air separation plants are not considered here.

In this study, the feed to the plant is assumed to be treated biogas and an oxygen-rich stream (95 mol% $O_2$ + 5 mol% $N_2$) both at 313 K and 1.3 bar. The total inlet methane flowrate is specified at 15,000 $Nm^3_{CH_4}\,h^{-1}$, which can represent 50 mol% to 70 mol% of the biogas composition with the rest being solely $CO_2$. Such biogas production volumes can only be achieved at large bioethanol plants, i.e., approximately 1,000,000 $m^3_{Ethanol}\,year^{-1}$ or by combining the output of two or more plants. A previous study on a medium-sized bioethanol plant producing 1320 $Nm^3\,h^{-1}_{CH_4}$ revealed a rather insufficient production volume in order to achieve decent economies of scale and dilute investment in expensive process equipment [5].

The main product is polymer-grade ethylene (0.9995 $mol_{C_2H_4}\,mol^{-1}$) and the resulting plant output is 4632 $t_{C_2H_4}\,year^{-1}$. The side products are a refinery-grade ethane stream and a methane-rich light off-gases stream. Both side streams are sold for additional revenue as detailed in Section 2.4.6.

The models described in the following sections are implemented in the process simulation software Aspen Plus and Aspen Custom Modeler v10. Further details on the models are provided in [25] and the Aspen Plus model files are made available in [26]. All process optimizations have been performed via a self-programmed Python interface to Aspen Plus [27] and by applying Differential Evolution as optimization algorithm [28].

## 2.1. Reaction Section Model

The OCM reaction is carried out in adiabatic Packed Bed Reactors (PBRs) with oxygen as the limiting reactant. Due to the high temperatures involved and high exothermicity of the reaction network, supplying heat and controlling temperature are difficult tasks. Therefore, industrial operation with PBR in isothermal regime is unrealistic and adiabatic regime is likely the only current option available with this reactor type [29]. To cope with low per-pass product yields and high temperature rises, two reactors in series with intermediate cooling and oxygen feed are adopted in this study.

The reactors are modeled using the Plug-Flow Reactor (PFR) model in Aspen Plus (RPlug) with kinetics available in literature for $La_2O_3$/CaO catalyst [7]. The reactor model implementation is validated by comparing simulation results to lab-scale isothermal experiments by [7]. The comparison in terms of ethylene yield (defined in Equation (11)) as a function of the contact time (relation between amount of catalyst and gas flow rate) for two different temperatures is shown in Figure 2. The simulated and experimental results are very comparable. The sharp bend occurring at 830 °C (1103 K) is due to oxygen extinction in the reactor.

$$Y_{C_2H_4} = \frac{\dot{N}^{out}_{C_2H_4} - \dot{N}^{in}_{C_2H_4}}{\frac{1}{2} \cdot \dot{N}^{in}_{CH_4}} \tag{11}$$

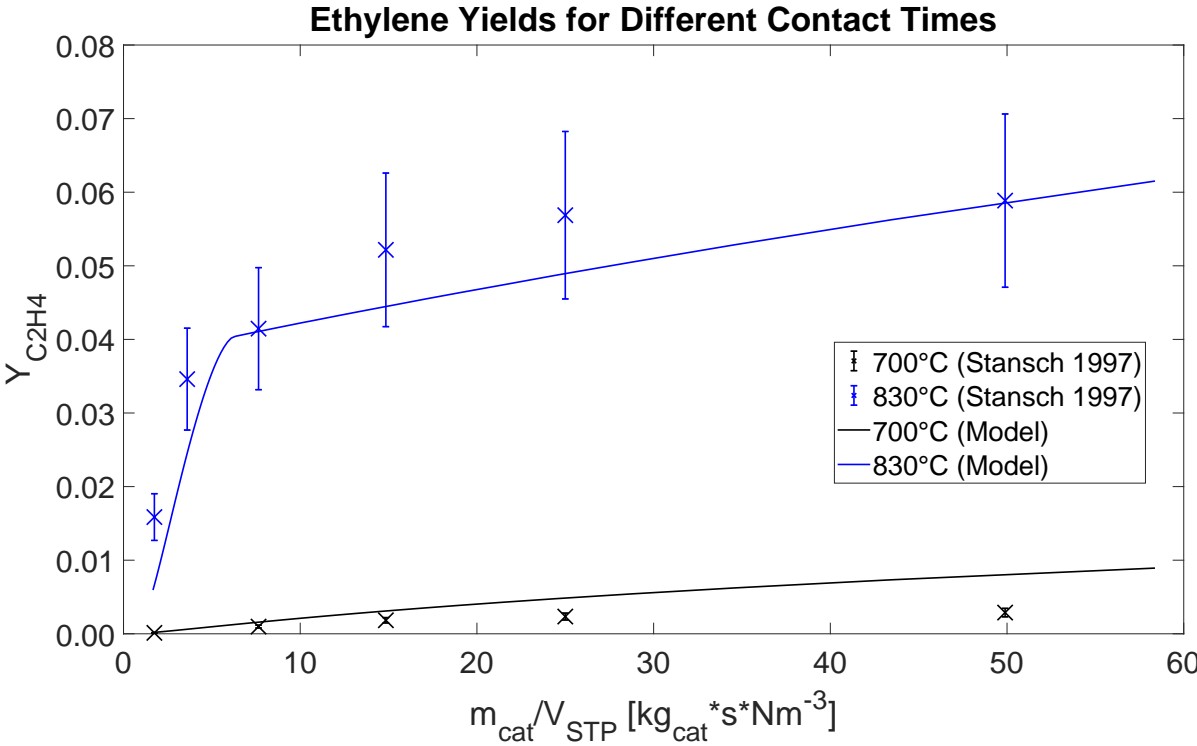

**Figure 2.** Product ($C_2H_4$) yield for different contact times obtained with an OCM-PBR operated isothermally at $700\,°C$ and $830\,°C$. Reactor model predictions compared to experimental data from [7]. Reproduced with permission from Penteado, Ph.D. Thesis, TUB, 2021 [25].

### 2.2. Carbon Dioxide Removal Section Model

Two competing process structures are considered for this section, i.e., a standalone amine-absorption process and a hybrid process that applies both Gas Separation Membranes (GSM) and amine absorption. Amine absorption is used in a wide variety of industrial processes for $CO_2$ removal. A common benchmark amine solution is Monoethanolamine (IUPAC: 2-aminoethan-1-ol) (MEA), which is usually employed in aqueous solutions of up to 30 wt%. The solution reacts promptly with $CO_2$ in the absorption column, but its regeneration in a secondary desorption column consumes a significant amount of energy (up to $5.0\,MJ\,kg_{CO_2}^{-1}$) [30]. The hybrid process consists in applying GSM, in this case polyimide polymeric membrane modules, to partially remove $CO_2$ in the upstream of the absorption column. This reduces the required amount of recirculating amine solution and, therefore the energy associated with its regeneration. However, the hybrid process may potentially require additional compression to drive separation through the membranes, higher capital investment in compressors, and lead to additional product (ethylene) losses through the membranes. To define the optimal process structure and operating conditions, a superstructure containing both alternatives is formulated to solve an economic optimization problem.

#### 2.2.1. Absorption Model

The process model of the Bio-OCM absorption section uses the Electrolyte Non-Random Two-Liquid (eNRTL) activity model [31,32] and the Peng-Robinson (PR) Equation of State (EoS) [33]. The columns are simulated using a conventional phase and chemical equilibrium approach through Aspen Plus' block RadFrac. The solubility of carbon dioxide in an aqueous solution of 30 wt% MEA predicted by the model is compared to experimental data by [34] in Figure 3. A very good agreement can be observed within the relevant range of process conditions.

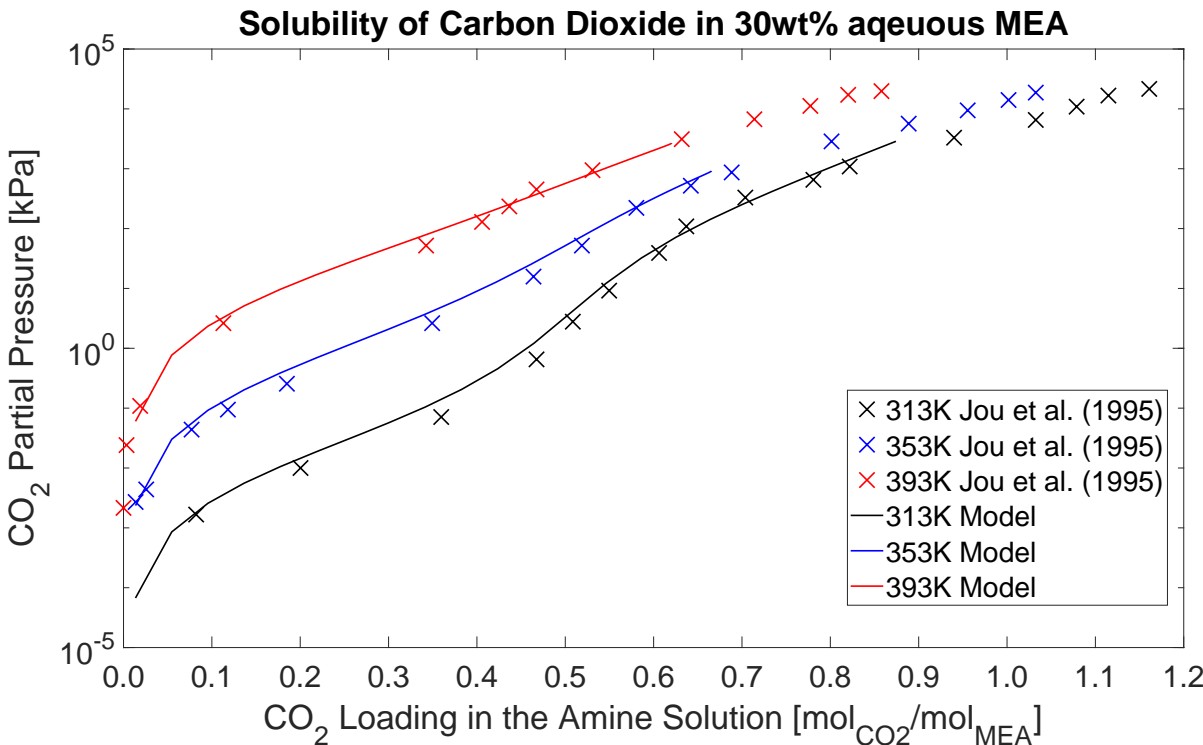

**Figure 3.** Carbon dioxide loading in 30 wt% MEA water solution at different partial pressures and temperatures. Comparison of values predicted by the developed model and experimental data from [34]. Reproduced with permission from Penteado, Ph.D. Thesis, TUB, 2021 [25].

The solubility of hydrocarbon gases in aqueous amine solutions is usually higher than that in pure water, which several authors refer to as salting-in effect [35]. Therefore, the parameters used to calculate the Henry constant of all gases in water and in MEA have been fitted to experimental data.

### 2.2.2. Gas Separation Membranes (GSM) Model

For this application, flat-sheet envelope-type membrane modules developed at Helmholtz-Zentrum Geesthacht are considered [36]. Among the previously tested membrane materials, a polyimide-based membrane has been selected for its high $CO_2$ selectivity towards several hydrocarbons [37]. For this specific application, selectivity is relatively more important than permeability, since ethylene is a valuable product and losses must be kept at minimum even if this implies large membrane areas. Since the membrane is selective for $CO_2$, its concentration increases in the permeate stream and reduces in the retentate stream.

For GSM, a previously published one-dimensional solution-diffusion model has been used [37]. The component's flux through the membrane is calculated as the product of the permeance and the driving force. The first is fitted to mini-plant experimental data [38], and the second is given by the difference in the component's fugacity on each side of the membrane calculated by the PR-EoS. The resulting Differential Algebraic Equation System (DAEs) is discretized by orthogonal collocation on finite elements, implemented in the software Aspen Custom Modeler, and exported as a custom unit operation into Aspen Plus.

### 2.3. Distillation Model

The final separation section contemplates two distillation columns, namely de-methanizer, which is responsible for recovering the light components ($CH_4$, $H_2$, CO, $N_2$), and $C_2$-splitter, which is responsible for $C_2H_4$ and $C_2H_6$ separation. Two different designs are compared on an economic basis for this section. The first one is herein called

traditional configuration because it has been previously adopted by several authors [39–41]. The traditional configuration uses only external refrigeration cycles to generate the cold utility required to condense the columns' top products. The petrochemical and natural gas processing industries, however, usually employ different schemes that expand a fraction of the hydrocarbon feed or products to produce "in-process" cold utility and reduce the load of external refrigeration cycles. To assess the potential of applying such schemes within a Bio-OCM process, a functional Recycle Split Vapor (RSV) distillation configuration is developed taking patent [42] as a basis and compared to the traditional distillation configuration.

The columns are modeled using Aspen Plus' RadFrac block under phase-equilibrium assumption. The PR-EoS with its original mixing rule and full set of binary parameters has been used to model this part of the process [33]. In both columns, constraints (design-specs in Aspen Plus) are used to achieve target purities and recoveries by manipulating input variables such as the reflux and boil-up ratios. In the demethanizer bottoms, methane contamination is set to ($1 \times 10^{-4}\,\mathrm{mol}_{CH_4}\,\mathrm{mol}^{-1}$), while ethylene recovery is set to 99%. In the top of the $C_2$-splitter, ethylene recovery is set to 99% while its purity is set to achieve polymer-grade, i.e., $0.9995\,\mathrm{mol}_{C_2H_4}\,\mathrm{mol}^{-1}$.

*2.4. Cost Models*

This section describes the cost models used to estimate variable costs (educts, side-products, and utilities) as well as fixed cost (equipment).

2.4.1. Total Annualized Cost

The total annualized cost per mass of product (bio-ethylene), i.e., $TAC_{C_2H_4}$ in $\mathrm{USD\,kg}_{C_2H_4}^{-1}$, is given by Equation (12). The $TAC_{C_2H_4}$ encompasses the utility cost and annualized equipment cost, while also considering product losses by placing the ethylene output mass flow in the denominator. The $TAC_{C_2H_4}$ is used in the process design stage as the objective function to be minimized within a process optimization problem or as the criteria to compare alternative process configurations. Throughout this study, a plant's operating life ($N$) of 30 years and an interest rate ($i_R$) of 10% have been used.

$$TAC_{C_2H_4} = \frac{1}{\dot{F}_{C_2H_4}^{mass}} \cdot \left( UtilityCost + EquipmentCost \cdot \left( \frac{i_R \cdot (i_R + 1)^N}{(1 + i_R)^N - 1} \right) \right) \qquad (12)$$

2.4.2. Ethylene Production Cost

For the economic evaluations described in Item Section 4, the total production cost of bio-ethylene, i.e., $c_{C_2H_4}$ in $\mathrm{USD\,kg}_{C_2H_4}^{-1}$, is calculated as per Equation (13). Besides utility cost and annualized equipment cost, the $c_{C_2H_4}$ also includes the cost of educts and revenues from side products. This is then divided by the output mass flow rate of ethylene. The equipment cost is annualized by the same formula as in Equation (12).

$$c_{C_2H_4} = \frac{\dot{c}_{Utility} + \dot{c}_{AnnualizedEquipment} + \dot{c}_{Educts} - \dot{c}_{SideProducts}}{\dot{F}_{C_2H_4}^{mass}} \qquad (13)$$

The calculated production cost for bio-ethylene ($c_{C_2H_4}$) is then compared to typical market values for fossil ethylene, which can range from $0.70\,\mathrm{USD\,kg}_{C_2H_4}^{-1}$ to $1.50\,\mathrm{USD\,kg}_{C_2H_4}^{-1}$ and fluctuate significantly with the oil price [43]. For the plant to be constructed, several other costs such as civil engineering, land and terrain, instrumentation and automation, electric engineering, operation and maintenance, taxes, depreciation, contingency, etc., would also incur. Environmental costs are also not taken into account in this economic calculations. Therefore, the bio-ethylene production cost calculated by Equation (13) must be significantly lower than ethylene's market value to justify the economic potential for a Bio-OCM plant. The calculation of the individual components of Equation (13) are detailed in the following sub-sections.

### 2.4.3. Monte Carlo Simulation

There is a large degree of uncertainty in all terms of Equation (13). In some cases, even small variations can significantly alter the outcome of the analysis. To deal with this, a Monte Carlo simulation is performed. Reasonable cost ranges, i.e., bounds, are given to each one of the cost components contained in Equation (13). The bio-ethylene production cost is then computed 10,000 times with random values between bounds being assigned for each cost component. This allows for a statistical interpretation of the results. For instance, one can compute in how many of these 10,000 samples the bio-ethylene production cost results lower than a target value, i.e., the probability that the bio-ethylene production cost is below a target value. The following Sections 2.4.4–2.4.6 discuss, for each term of Equation (13), the considered ranges/bounds for each variable in the Monte Carlo simulation. The randomize function of Microsoft Office Excel has been used to generate the random values for the Monte Carlo sampling.

### 2.4.4. Utility Cost

The Bio-OCM consumes electricity and Light Pressure Steam (LPS) and Medium Pressure Steam (MPS) for heating, while exporting High Pressure Steam (HPS) produced by using the reactions' heat release. The electricity cost for small-medium enterprises in Brazil, which is around $0.126\,USD\,kWh^{-1}$ ($35\,USD\,GJ^{-1}$) [44], is adopted. This is additionally employed to modify the cost of low, medium, and high-pressure steam from their default values in Aspen Plus. The conditions and cost for all applied utilities are summarized in Table 1. For the Monte Carlo simulation, the ranges are set to $\pm30\%$ of the nominal values in Table 1.

**Table 1.** Conditions and cost of utilities used/generated in the Bio-OCM plant.

| Utility Type | Condition | Cost of Energy in USD GJ$^{-1}$ |
|---|---|---|
| Electricity | - | 35.0 |
| LPS | 2.3 bar/125 °C | 3.09 |
| MPS | 8.9 bar/175 °C | 3.58 |
| HPS | 39.7 bar/250 °C | 4.05 |
| Cooling Water | 20–25 °C | 0.212 |

### 2.4.5. Equipment Cost

Sizing and costing of equipment is performed using activated economics in Aspen Plus, which essentially transfers simulation data into Aspen Process Economic Analyzer (APEA) software. APEA contains automated procedures for equipment sizing and costing based on an updated data-bank. In Item Section 4.3, a location factor of 1.4 is applied to correct the installed equipment cost estimated by APEA, as suggested in other studies for chemical plants based in Brazil [45].

Cost estimations in conceptual and basic engineering are typically assumed to have a $\pm30$–$50\%$ error margin. Given the difficulties in estimating cost for royalties to be paid, notably for the reactor and catalyst technologies, a range of $-30\%$ to $+50\%$ is considered in the Monte Carlo simulation as a simple and conservative solution.

### 2.4.6. Cost of Educts and Side Products
Biogas

The cost for producing biogas depends on a handful of parameters such as the substrate, transportation, plant capacity, and the amount and nature of contaminants. A general study reports ranges of $0.03\,USD\,Nm^{-3}$ to $0.05\,USD\,Nm^{-3}$ [46]. Specifically for biogas derived from vinasse AD, estimates in the range of $0.022\,USD\,Nm^{-3}$ to $0.038\,USD\,Nm^{-3}$ [47] and $0.0525\,USD\,Nm^{-3}$ [48] have been identified. For the Monte Carlo simulation, the entire range of $0.022\,USD\,Nm^{-3}$ to $0.0525\,USD\,Nm^{-3}$ has been adopted.

Oxygen

Oxygen is assumed to be provided by an adjacent air separation unit, which is not included in the model. Oxygen-rich streams can be obtained industrially from air via cryogenic distillation or Pressure-Swing Adsorption (PSA). Since a high oxygen purity it not required for this application, a 95 mol% oxygen stream is assumed with the contamination consisting only of inert nitrogen. For the subsequent Monte Carlo simulation, the assumed oxygen delivery price range is $0.04\,\mathrm{USD\,kg_{O_2}^{-1}}$ to $0.10\,\mathrm{USD\,kg_{O_2}^{-1}}$ [49].

Ethane

Ethane is considered to be sold as a side-product with a price in the range of $0.0467\,\mathrm{USD\,kg_{C_2H_6}^{-1}}$ to $0.0686\,\mathrm{USD\,kg_{C_2H_6}^{-1}}$ based on [43].

Light Off-Gas

The main side product of the Bio-OCM plant is a light gas stream, which contains mostly the un-reacted $CH_4$ ($\approx$90 mol%) along with minor amounts of $H_2$, CO, $N_2$, and trace amounts of $C_2H_4$ and $C_2H_6$. A sales price close to that of pipeline natural gas in Brazil, which ranges from $5.00\,\mathrm{USD\,GJ_{HHV}^{-1}}$ to $6.91\,\mathrm{USD\,GJ_{HHV}^{-1}}$ [44,45], is assumed. For the subsequent Monte Carlo simulation, a sales price range between $2.0\,\mathrm{USD\,GJ_{HHV}^{-1}}$ to $5.0\,\mathrm{USD\,GJ_{HHV}^{-1}}$ is assigned to the light off-gas stream. Natural gas specification in Brazil is given by a resolution or standard ANP No. 16-2008 by the National Agency for Petroleum, Natural Gas and Biofuels. Table 2 compares the specification of pipeline natural gas in Brazil with those of the lights stream resulting from this simulation study. As it can be seen, the lights stream almost matches the required specifications for pipeline-quality natural gas from the standard ANP 16-2008, thus justifying a similar sales price. For additional reference, characteristics of other conventional and unconventional gaseous fuels are reported. These are based on data from [50], which include biogas from sewage sludge and gas obtained from catalytic pyrolysis of high-density poly-(ethylene), i.e., PPG.

**Table 2.** Characteristics of the light gas steam side-product of the Bio-OCM process in comparison with pipeline natural gas specifications from Brazilian standard ANP 16-2008 and other fuel gases.

| Fuel Gas Characteristic | Lights Stream (Bio-OCM Off-Gas) | Natural Gas Brazil (ANP 16-2008) | Biogas [50] | PPG [50] | Units |
|---|---|---|---|---|---|
| Higher heating value [1] | 35.17 | 35.0–43.0 | 24.32 | 115.57 | $\mathrm{MJ\,m^{-3}}$ |
| Wobbe index [1] | 46.13 | 46.5–53.5 | 25.64 | 76.71 | $\mathrm{MJ\,m^{-3}}$ |
| Methane number | 591.16 | $\geq$65 | | | - |
| Methane content | 90.4 | $\geq$85.0 | 64.22 | 1.36 | mol% |
| Inert concent ($N_2$+$CO_2$) | 1.3 | $\leq$6.0 | 35.4 | 0.8 | mol% |
| Hydrocarbon dew point [2] | −89 | $\leq$0 | | | °C |
| Cost Range | 2.0–5.0 | 5.00–6.91 | | | $\mathrm{USD\,GJ_{HHV}^{-1}}$ |

[1] At 288 K and 101.325 kPa. [2] At 4.5 MPa.

## 3. Process Design Results

This section provides results regarding the selected process structure and operating conditions. The full set of material and energy balances for the flowsheets is provided in [25].

### 3.1. Reaction Section Design

The simulation flowsheet for the reaction section of the Bio-OCM process is shown on Figure 4. The biogas feed is simulated by mixing a pure $CH_4$ stream named CH4 and a pure $CO_2$ stream named CO2. This is done to investigate the effect of different inlet $CO_2$ concentrations to the reaction performance. The biogas stream is mixed with the 95 mol% $O_2$ stream named O2-1 in mixer MIX-1 and passes through two pre-heating exchangers E-01 and E-02 reaching the feed temperature for the first reactor R-01. The furnace blocks

F-01 and F-02 are required for start-up only and the system is auto-thermal in steady-state. The product gas of R-01 is cooled in a heat exchanger E-03 by generating HPS, mixed with a second $O_2$ stream named O2-2, and further cooled in E-01 to the feed temperature of the second reactor R-02. The product gas of R-02 is again cooled in a heat-exchanger E-04 by generating HPS and further heat is recovered in E-02.

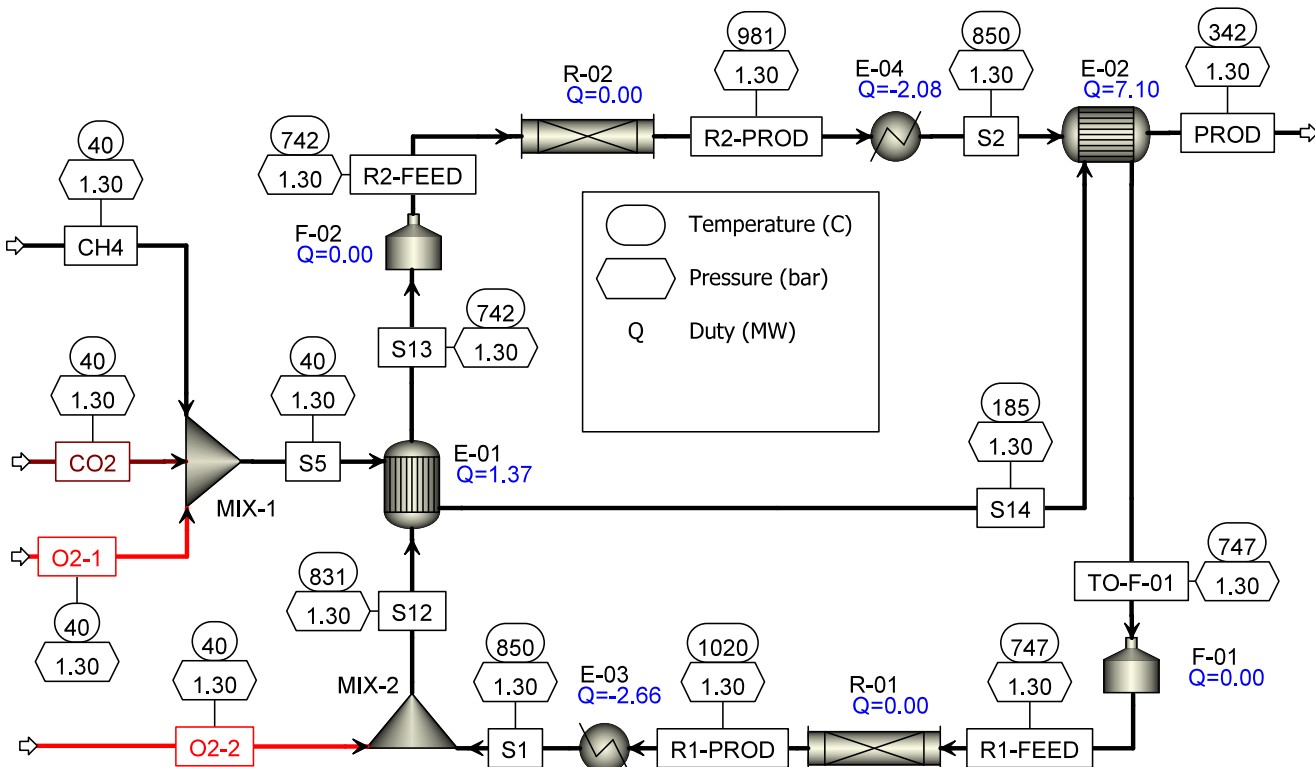

**Figure 4.** Process simulation with optimal operating conditions for the reaction section of the Bio-OCM plant carried out in Aspen Plus software. Black lines (process streams) and red lines (oxygen streams). Reproduced with permission from Penteado, Ph.D. Thesis, TUB, 2021 [25].

The process conditions are determined by means of a mathematical optimization targeting the maximization of the combined ethylene and ethane yield ($C_2$ yield) as given by Equation (14). The decision variables are the feed temperatures to each reactor, the amount of $CO_2$ fed via the stream named $CO_2$, the amounts of $O_2$ fed via streams named O2-1 and O2-2, and the amount of catalyst, i.e., contact time, in each reactor. The resulting $C_2$ yield is 16.12%, while methane conversion is 24.9% and selectivity towards $C_2$ products is 64.7%. The relatively low conversion means that a significant amount of $CH_4$ is still available in the reaction product gas. The resulting $C_2H_4$ to $C_2H_6$ ratio is 1.5, thus a significant amount of ethane is also produced as side product. The obtained performance is not exhilarating. In fact, $C_2$ yields as high as 24.2% have been previously obtained in a mini-plant set-up with packed-bed membrane reactors [51]. However, the results are well in line with current scientific publications and patents that also adopt adiabatic operation with PBR [29,42]. Thus, it provides a realistic performance in terms of what could be achieved on an industrial-scale implementation to date.

$$Y_{C_2} = \frac{\dot{N}_{C_2H_4}^{PROD} + \dot{N}_{C_2H_6}^{PROD}}{\frac{1}{2} \cdot \dot{N}_{CH_4}^{CH4}} \tag{14}$$

The optimal amount of $CO_2$ in the biogas feed is close to its upper limit, which implies a biogas feed with approximately 50 mol% $CO_2$. Therefore, there is no need for

an upstream $CO_2$ removal step as the $CO_2$ dilution has a positive effect to the reaction performance. The inlet $O_2$ flow rates lead to methane to oxygen ratios of $8.4 \, \text{mol}_{CH_4} \, \text{mol}_{O_2}^{-1}$ and $10.0 \, \text{mol}_{CH_4} \, \text{mol}_{O_2}^{-1}$ in R-01 and R-02, respectively. These are significantly higher than the stoichiometric ratio of $2 \, \text{mol}_{CH_4} \, \text{mol}_{O_2}^{-1}$, but this is expected for adiabatic operation as high oxygen availability leads to undesired combustion reactions coupled with strong heat release.

The optimal feed temperatures are 1020 K and 1015 K for R-01 and R-02, respectively. These are high inlet temperatures as adiabatic operation also leads to a high temperature rise along the reactors. As temperatures can reach above 1273 K, a reactor with a refractory lining is required and thermal stability of the catalyst may become an issue for long-term operation. Low-temperature OCM catalysts that are active below 973 K offer great potential to simplify operation and achieve higher yields in adiabatic operation, but these are still new and underdeveloped [52]. In this sense, there is an urge to re-evaluate the performance of several OCM catalysts in adiabatic regime, given that this remains the best alternative for industrial implementation [29]. Specifically for biogas or any $CO_2$-diluted process feed, it is also essential that future studies investigate the influence of $CO_2$ on the reaction and catalyst on an experimental level and also revise current kinetic models so that they cover this range of operating conditions. Therefore, there are still technical challenges that need to be addressed prior to industrial implementation.

The reaction section produces HPS as its only utility, which is exported to generate a revenue (negative cost) of 605,638 USD year$^{-1}$. Figure 5 shows the total equipment cost estimated at $2.25 \times 10^6$ USD divided into equipment categories. The equipment cost is quite balanced between heat exchangers, furnaces, and the reactors, which are tubular carbon-steel vessels with refractory lining. Additional incurring costs, e.g., royalties for catalyst and reactor technology, are not estimated but assumed to be covered by the conservative $-30\%$ to $+50\%$ equipment cost range used in the Monte Carlo simulation later in Item Section 4.5.

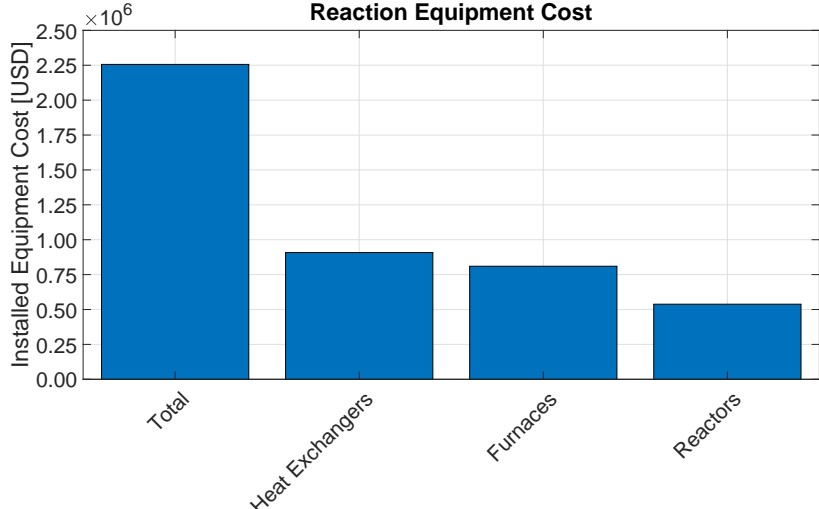

**Figure 5.** Estimated cost of equipment installed in the reaction section of the Bio-OCM plant. Reproduced with permission from Penteado, Ph.D. Thesis, TUB, 2021 [25].

### 3.2. Carbon Dioxide Removal Section Design

The simulation flowsheet containing the superstructure for the $CO_2$ removal section is given in Figure 6. A splitter block (SPL-202) and a by-pass stream (M-BYPASS) allow for the simulation of both configurations, i.e., standalone absorption and hybrid membrane-absorption. The product gas from the reaction section is fed to a direct contact cooler or quench column (C-201), in counter-current with a recirculating cooling water stream. The cooled gas stream is compressed in one or two stages (K-201 and K-202) with inter-stage

water cooling (E-201 and E-202) and condensate removal drums (D-201 and D-202). If the membrane section is used, the gas is dried (X-201), further compresssed (K-301), cooled (E-307), and fed to the first membrane module (M-301). The $CO_2$-rich permeate stream may be purged (SPL-301) or re-compressed (K-302) and forwarded to a second membrane module (M-302) in order to enhance ethylene recovery. The retentate stream from M-301 feeds three absorption stages in series (C-301, C-302, and C-303) counter-current to the recirculated MEA solution. The latter is distributed to the three stages at different ratios via a splitter block (SPL-302) and the bottom of each absorption stage may be cooled to shift the chemical equilibrium towards absorption. The rich or loaded amine solution is flashed to near atmospheric pressure (D-301), pressurized (P-301), cooled by heat-recovery (E-304), and enters the desorption or amine regeneration column (C-304). Amine regeneration is accomplished by MPS, producing a nearly pure $CO_2$ stream (CO2-OUT) at the top and a lean amine stream at the bottom. The lean amine stream is again pressurized (P-302), heated by heat-recovery (E-304), and enters the mixer (MIX-302). A make-up stream is added to MIX-302 to replenish evaporative losses of amine and water in the process. The output of MIX-302 is cooled to 45 °C (318 K) and generates the recycle stream to the absorption (LEAN-IN). The $CO_2$-free top product stream of C-303 is further cooled, passes through a knock-out drum (D-303), is dried (X-301), compressed (K-303, K-304, and K-305), and forwarded to the distillation section (TO-DIST). A 97% removal of the inlet $CO_2$ is fixed and ensured by manipulating the amine re-circulation flow rate via the tear-stream LEAN-IN. The final $CO_2$ removal must be performed by a caustic wash, which has not been included in the model for simplicity.

The process structure or configuration and operating conditions for the $CO_2$ removal section are determined by a superstructure optimization targeting the minimization of the total annualized cost per mass of ethylene output, i.e., $TAC_{C_2H_4}$ as given in Item Section 2.4.1. The main decision is whether to use a standalone absorption configuration or a hybrid membrane-absorption configuration, but the full set of 13 decision variables include the pressure ratios in compressors K-201, K-202, K-203, K-303, and K-304 given in $bar\,bar^{-1}$; the membrane areas in membrane modules M-301 and M-302 given in $m^2$; the split fractions in splitters SPL-202, SPL-301, SPL-302; the lean loading for the recirculating amine solution given in $mol_{CO_2}\,mol_{MEA}^{-1}$; and the number of stages and feed stage in C-304. The discharge pressure in K-305 is fixed at 30 bar, which is the required pressure for the downstream distillation.

The optimal process configuration is found to be standalone absorption at low pressure (3.13 bar). These are the conditions shown in Figure 6. For comparison purposes, a second optimization is run by enforcing the use of GSM, i.e., the hybrid membrane-absorption configuration. The comparison, in terms of utility cost rate per utility category and equipment cost per equipment category are shown in Figures 7 and 8, respectively. The use of GSM can reduce the specific amine regeneration energy from $3.04\,MJ\,kg_{CO_2}^{-1}$ down to $2.50\,MJ\,kg_{CO_2}^{-1}$, which leads to a reduced cost rate for MPS. However, the steam cost savings are unjustified as the most significant contributions to the $TAC_{C_2H_4}$ actually stem from electricity cost rate and investment cost for compressors. The use of GSM requires additional compression in the upstream (K-201, K-202, and K-301) to provide enough driving force for permeation-based separation, which increases electricity cost rate and compressor equipment cost. On the other hand, standalone absorption can be carried out at lower pressures, which means that the downstream compressors (K-303, K-304, and K-305) take up the majority of the compression duty after $CO_2$ has been removed and the total gas flow rate to be compressed is significantly lower. Because of these, the hybrid membrane-absorption configuration has a 6.8% higher utility cost rate and a 17.7% higher equipment cost.

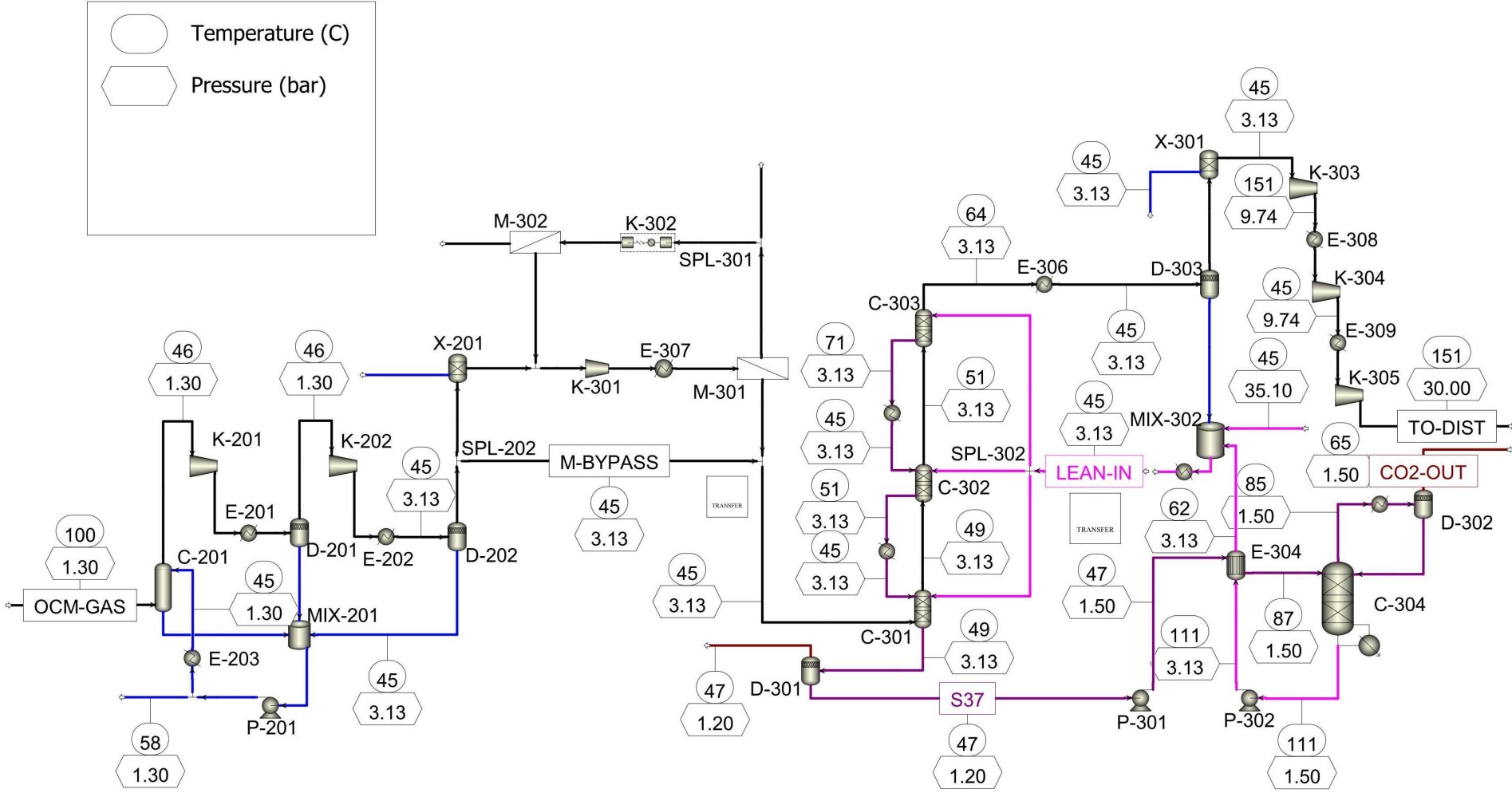

**Figure 6.** Process simulation with optimal process structure and operating conditions for the CO$_2$ removal section of the Bio-OCM plant using only absorption. Reproduced with permission from Penteado, Ph.D. Thesis, TUB, 2021 [25].

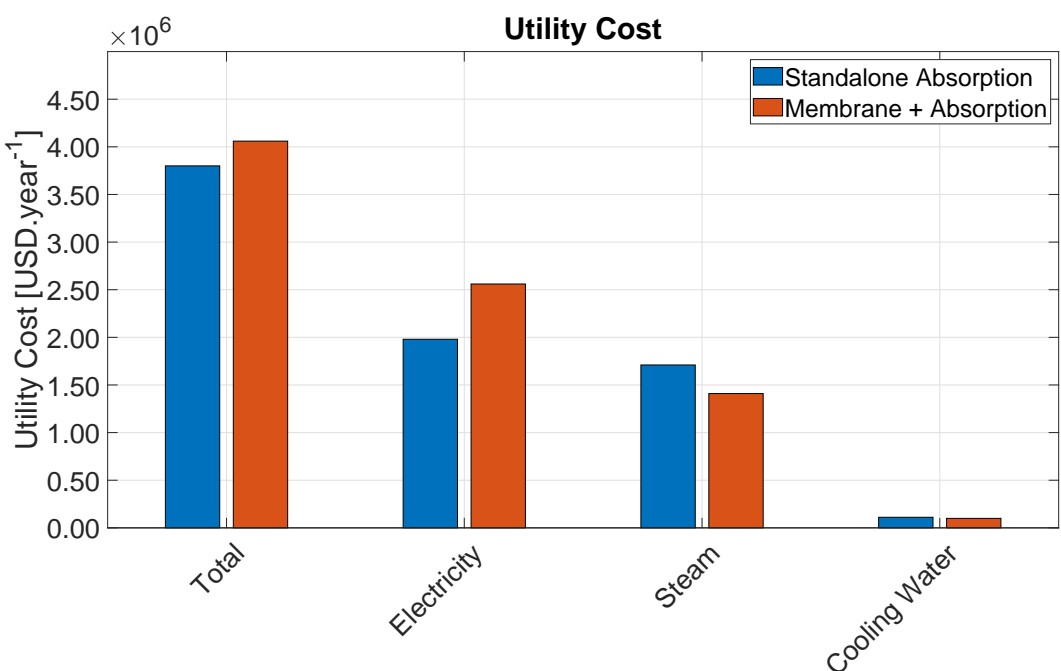

**Figure 7.** Cost rates of different utility in the $CO_2$ removal section for the two compared process configurations. Reproduced with permission from Penteado, Ph.D. Thesis, TUB, 2021 [25].

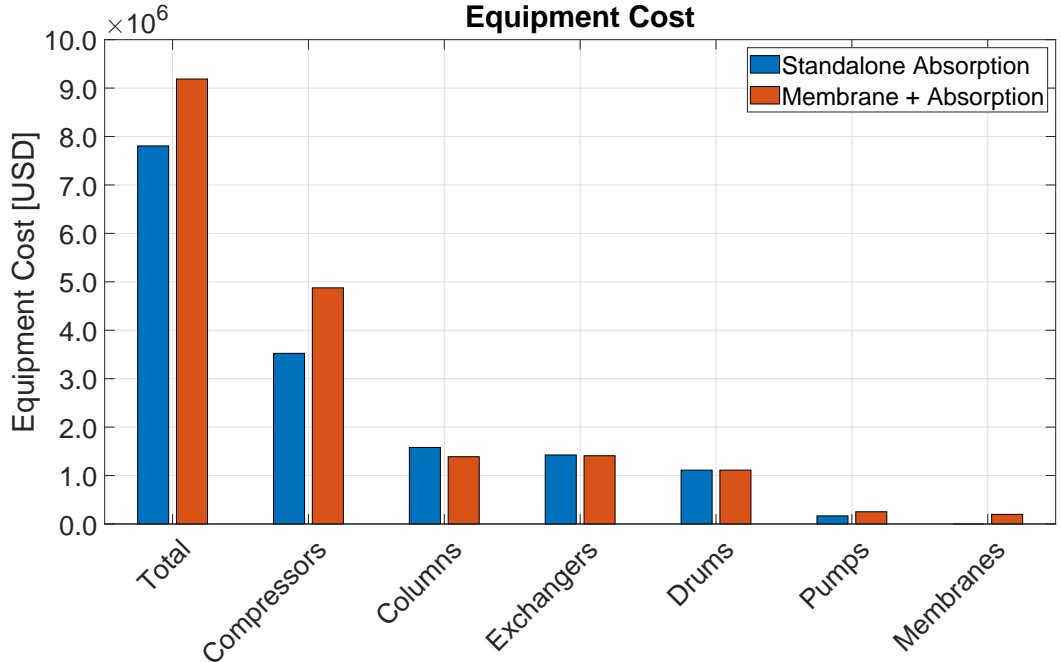

**Figure 8.** Cost of equipment installed in the $CO_2$ removal section for the two compared process configurations. Reproduced with permission from Penteado, Ph.D. Thesis, TUB, 2021 [25].

The product (ethylene) loss is also an important factor for computing the $TAC_{C_2H_4}$. Ethylene losses resulting from the simulations are 2.1% for the standalone absorption configuration and 7.3% for the hybrid configuration. This is because GSM present a lower $C_2H_4$ selectivity towards $CO_2$ than amine-absorption. In the hybrid configuration, some ethylene is purged out of the system in splitter SPL-301. In this configuration, it is possible to increase ethylene recovery by increasing the membrane feed pressure and/or by using a

higher recycle to M-302. However, both of these options lead to increased electricity cost rates and additional investment in process compressors. The adopted low pressure for the standalone absorption configuration also minimizes ethylene losses by reducing its physical absorption by the MEA solution.

Overall, the standalone absorption configuration outperforms the hybrid configuration regarding utility cost, equipment cost, and product recovery. The obtained $TAC_{C_2H_4}$ are $1.00\,\mathrm{USD\,kg_{C_2H_4}^{-1}}$ and $1.15\,\mathrm{USD\,kg_{C_2H_4}^{-1}}$ for the standalone absorption and hybrid configurations, respectively. Therefore, it can be concluded that the use of GSM is unlikely to bring any significant cost-savings to the Bio-OCM process.

### 3.3. Distillation Section Design

Figure 9 depicts the simulation flowsheet of the traditional distillation configuration. In this configuration, refrigeration fluids produced externally are applied in the condensers of both demethanizer and $C_2$-splitter columns. Due to the very low temperatures required in the top of the demethanizer column ($\approx$173 K), a 3-stage refrigeration cascade employing methane (R-50), ethylene (R-1150), and propylene (R-1270) is required. This refrigeration system is included as a sub-flowsheet (HIERARCHY block in Aspen Plus), which is represented by the squared block named REF in Figure 9 and fully shown in Figure 10. Methane at $-110\,°C$ (163 K) is employed in the demethanizer condenser, whereas ethylene at $-60\,°C$ (213 K) is employed in the $C_2$-Splitter condenser.

The simulation flowsheet for the RSV distillation configuration is shown in Figure 11. The feed gas for the distillation section is assumed to be free of $CO_2$, dry, and delivered at 3.13 bar and 318 K. The gas is initially compressed (K-401, K-402, K-403) to 30 bar and is pre-cooled in a multi-stream plate-fin heat-exchanger (MHX1), i.e., cold-box. The pre-cooled gas stream is flashed and fed to the demethanizer (C-401). The top product is splitted into two fractions. The stream TO-EXP is expanded (EXP-401), used as cold utility in both MHX2 and MHX1, and generates the off-gas side-product stream (LIGHTS). The stream TO-COMP can optionally be further compressed (K-405), but this compressor is by-passed in the final design to save capital expenditure. Stream S18 is partially condensed in MHX2, then flashed (D-403) to generate the demethanizer's reflux stream (C1-REFLX). The demethanizer's bottom stream (C2+) feeds the $C_2$-splitter (C-402) after being flashed to 8.7 bar to increase ethylene and ethane relative volatility without significantly lowering the mixture's dew point. Ethane is removed as the bottom product (C2-BOT) of the $C_2$-splitter. The stream coming at the top (C2-TOP) is heated-up in MHX3, compressed (K-404A), and splitted (C2-SPL). The stream TO-MHX3 is cooled down (MHX3), partially condensed in the $C_2$-splitter's reboiler (C-402REB), and generates the $C_2$-splitter's reflux stream (C2-REFLX). The stream S7 is further compressed (K-404B and K404C) and generates the main ethylene product stream (C2H4).

This configuration still requires external refrigeration utility. A refrigeration cascade applying only two stages with ethylene (R-1150) and propylene (R-1270) can be used and energy consumption is reduced significantly. In the RSV configuration, external refrigeration is only provided for pre-cooling (MHX1) by propylene at 227 K ($-46\,°C$) and ethylene at 188 K ($-85\,°C$). In Figure 11, this is again represented by a squared HIERARCHY block named REF, which contains the external refrigeration system sub-flowsheet shown in detail in Figure 12. In some cases, a single streams heaters and a single-stream cooler with opposite heat duties are employed to simulate two-stream heat-exchangers, i.e., MHX1-C2 and MHX1-C3 simulate MHX1 (cold-box in Figure 11), while E-504A and E-504B simulate the ethylene condenser/propylene evaporator. For cost calculations, only one heat-exchanger is considered.

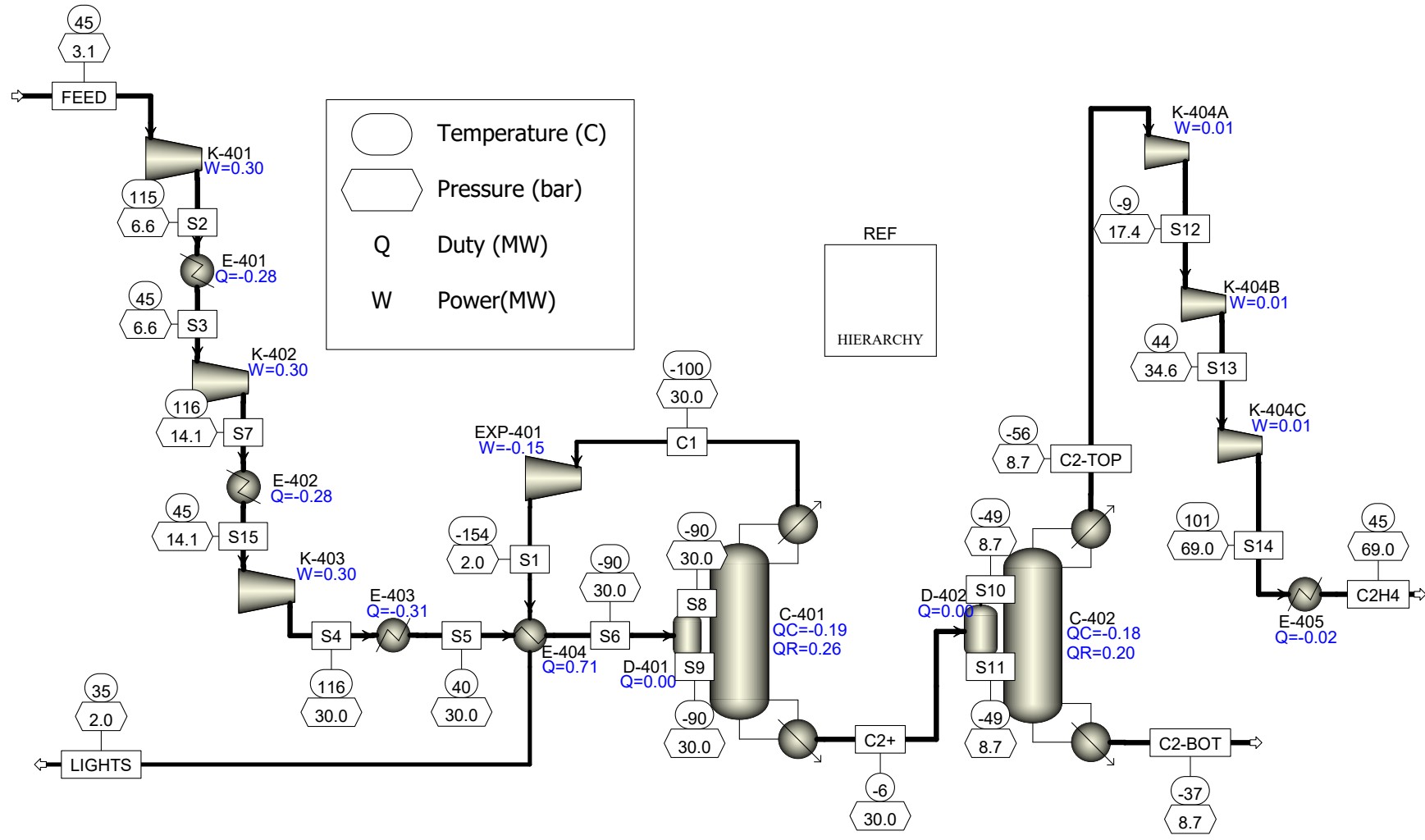

**Figure 9.** Process simulation of the traditional distillation configuration implemented in Aspen Plus software. Reproduced with permission from Penteado, Ph.D. Thesis, TUB, 2021 [25].

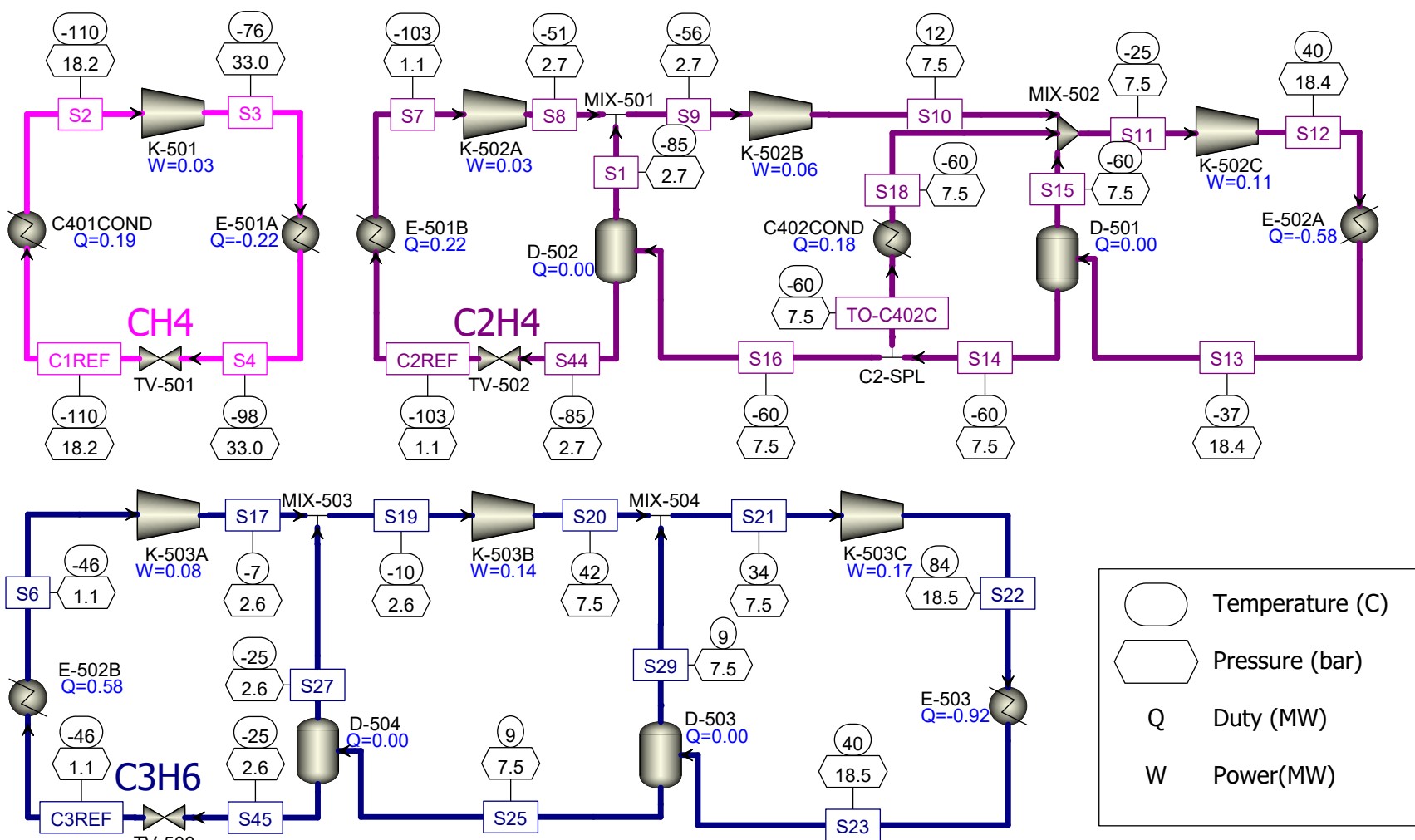

**Figure 10.** Process simulation of the external refrigeration system of the traditional distillation configuration implemented in Aspen Plus software. Reproduced with permission from Penteado, Ph.D. Thesis, TUB, 2021 [25].

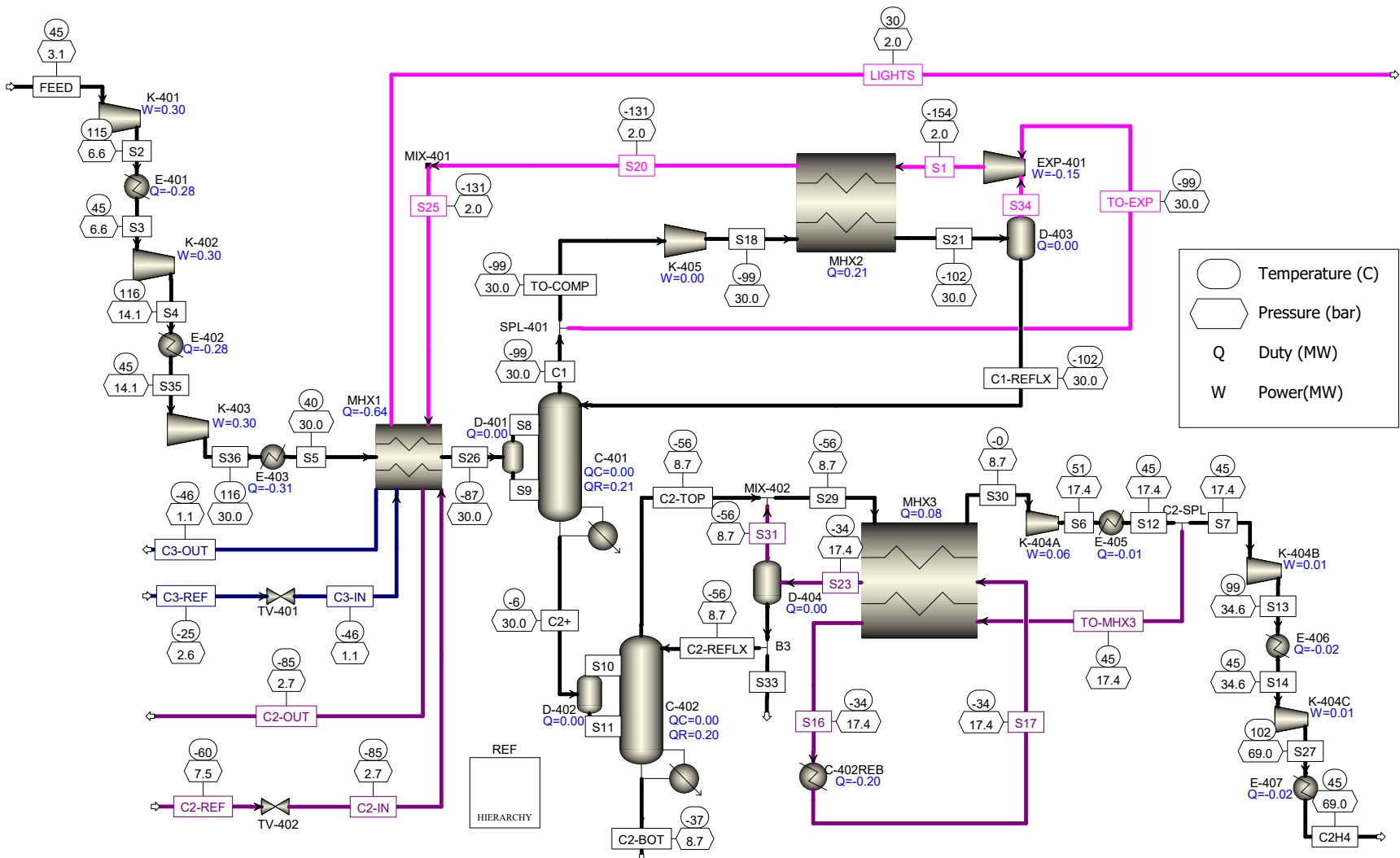

**Figure 11.** Process simulation of the RSV distillation configuration implemented in Aspen Plus software. Reproduced with permission from Penteado, Ph.D. Thesis, TUB, 2021 [25].

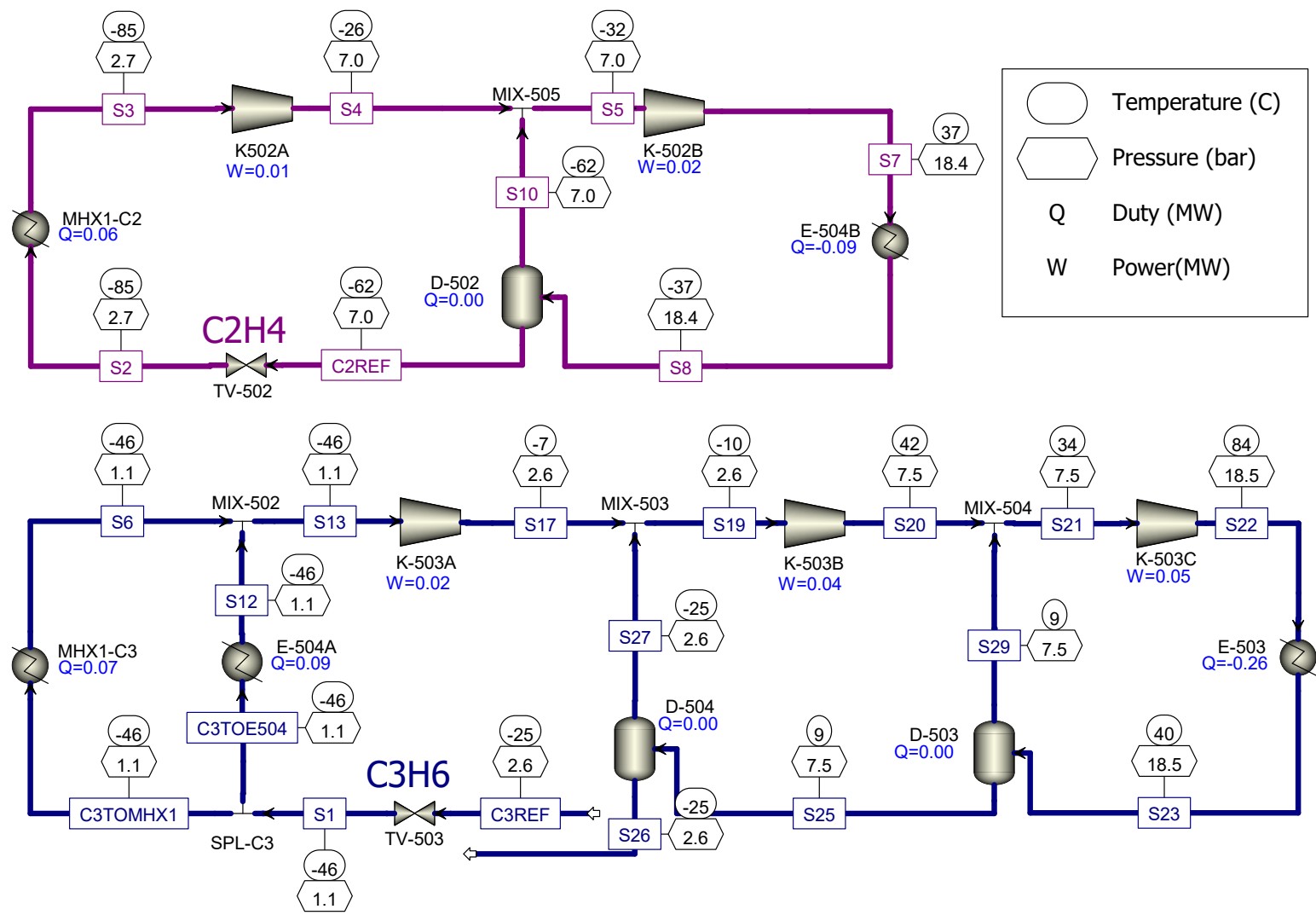

**Figure 12.** Process simulation of the external refrigeration system of the RSV distillation configuration implemented in Aspen Plus software. Reproduced with permission from Penteado, Ph.D. Thesis, TUB, 2021 [25].

Figure 13 shows the resulting cost rates for the different utilities employed in the traditional and RSV distillation configurations divided into different categories. All utility (electricity, steam, and cooling water) required to run the external refrigeration systems are grouped into the refrigeration utility category. The proposed RSV distillation configuration leads to a slightly higher process electricity cost rate, but can drastically reduce the refrigeration utility cost rate. The refrigeration utility cost rate is, in both cases, comprised mostly of electricity to run the refrigeration compressors. This configurations enables a better match of heating and cooling loads. The lowest temperature cooling duty at the demethanizer condenser is realized by the expanded demethanizer top stream (in-process generated utility), while the higher temperature cooling duty, i.e., feed pre-cooling, is realized by external refrigeration. In the traditional scheme, the opposite occurs. Overall, the newly developed RSV distillation configuration allows for an approximate 31% reduction in the total utility cost rate, i.e., $1.1 \times 10^6$ USD year$^{-1}$ vs. $1.6 \times 10^6$ USD year$-1$.

Figure 14 details the cost of different equipment installed in the traditional and RSV distillation confifugrations. All equippment (compressors, heat exchangers, and drums) present in the external refrigeration systems are grouped into the Refrigeration category. The proposed RSV distillation configuration requires slightly higher investments in process compressors and heat exchangers, but allows for a significant reduction in refrigeration equipment cost. The refrigeration equipment cost is, in both cases, comprised mostly of investment cost for compressors. The cost saving is achieved mostly because the RSV configuration removes the R-50 (methane) refrigeration compressor and the lowest temperature R-1150 (ethylene) refrigeration compressor. These are both necessary costly equipment in the traditional configuration (K-501 and K-502A in Figure 10). In total, the equipment costs are $13.8 \times 10^6$ USD and $11.6 \times 10^6$ USD for the traditional and RSV distillation configurations, respectively, so that a 16% equipment cost reduction has been achieved.

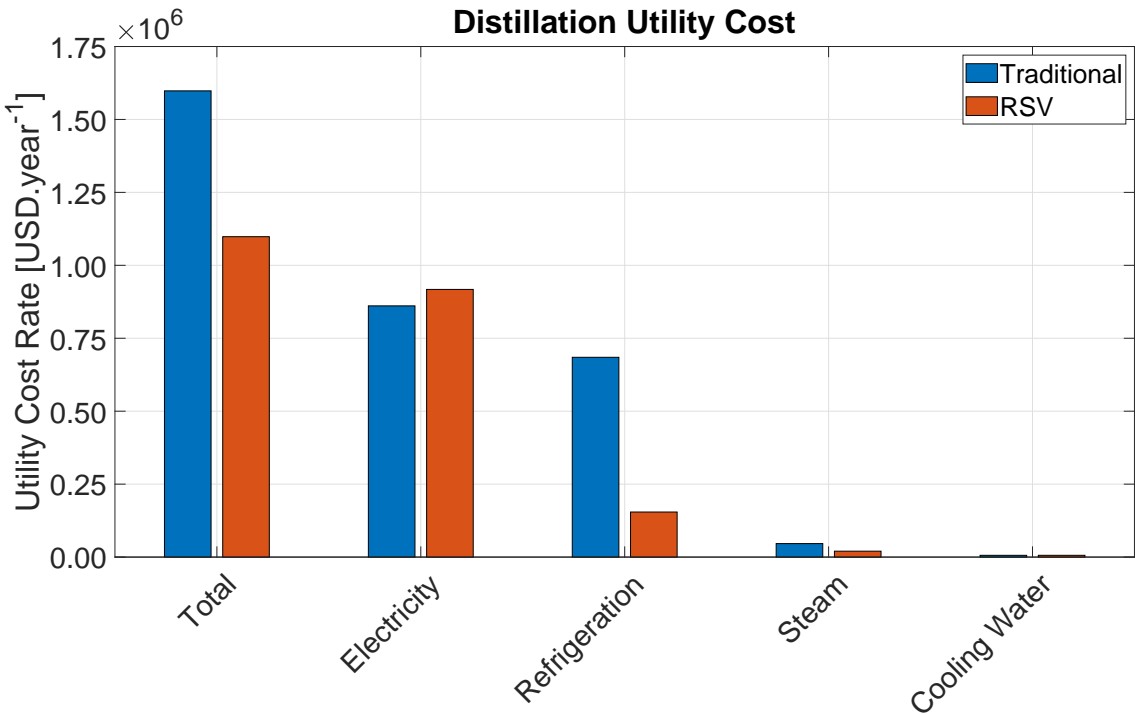

**Figure 13.** Cost rates of different utility in the distillation section for the two compared process configurations. Reproduced with permission from Penteado, Ph.D. Thesis, TUB, 2021 [25].

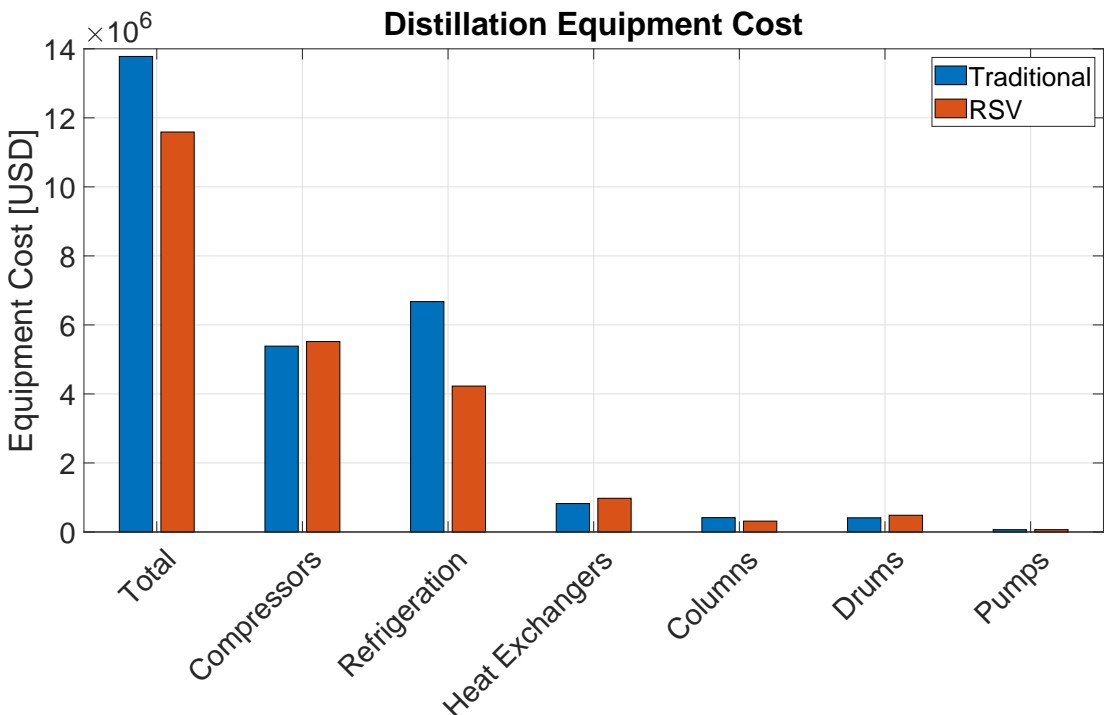

**Figure 14.** Cost of different equipment installed in the distillation section for the two compared process configurations. Reproduced with permission from Penteado, Ph.D. Thesis, TUB, 2021 [25].

In practice, the RSV distillation configuration may present additional challenges not addressed in this study, such as a more complex process control structure, longer start-up times, and less operational flexibility. However, it becomes clear that applying this configuration can lead to significant reduction in the ethylene production cost. By applying Equation (12), the computed $TAC_{C_2H_4}$ are 0.661 USD $kg_{C_2H_4}^{-1}$ and 0.503 USD $kg_{C_2H_4}^{-1}$ for the traditional and RSV distillation configurations, respectively. Thus, a 24% cost reduction is achieved.

## 4. Economic Evaluation

The optimal process configuration for the Bio-OCM plant led to a reaction section employing two adiabatically operated PBR in series, an amine-based standalone absorption process to remove carbon dioxide, and a low-temperature hydrocarbon distillation process applying a RSV configuration to reduce refrigeration cost. The Bio-OCM plant consumes biogas and oxygen as the main educts. The main product is bio-ethylene, while ethane and light gases are also obtained as side-products. In terms of utilities, electricity as well as low and medium pressure steam are consumed, while high pressure steam is exported.

### 4.1. Cost of Educts and Side Products

Table 3 summarizes all educts and products of the Bio-OCM process in terms of flow rates, assumed price ranges, and resulting cash flow rates. The yearly cash flows are the products of the flow rates of each educt/product and their respective price ranges. Worst-case cash flows occur with educt cost at upper limit and side-product prices at lower limits, respectively. In the best-case scenario, the contrary occurs. In the average scenario, the arithmetic mean between lower and upper bounds are taken for reference.

**Table 3.** Summary of all educts and side products of the Bio-OCM plant.

| Educt or Side-Product | Flow Rate | Flow Rate Units | Price Range | | Price Unit | Cash Flows in kUSD Year$^{-1}$ | | |
|---|---|---|---|---|---|---|---|---|
| | | | | | | Worst | Average | Best |
| Biogas | $131.4 \times 10^6$ | $Nm^3\,year^{-1}$ | 0.022 | 0.0525 | $USD\,Nm^{-3}$ | −6898 | −4895 | −2891 |
| Oxygen | 22.8 | $t\,year^{-1}$ | 40 | 100 | $USD\,t^{-1}$ | −2.3 | −1.6 | −0.9 |
| Ethane | 3342 | $t\,year^{-1}$ | 46.7 | 68.6 | $USD\,t^{-1}$ | 156.2 | 192.7 | 229.3 |
| Lights | 48,782 | $t\,year^{-1}$ | 2.0 | 5.0 | [1] $USD\,GJ_{HHV}^{-1}$ | 4818 | 8430 | 12,043 |
| TOTAL | | | | | | −1927 | 3727 | 9381 |

[1] $HHV_{gas}$ = 49.4 MJ kg$^{-1}$ = 35.17 MJ m$^{-3}$ @ 288 K & 101.325 kPa.

In terms of educt cash flows, biogas has the highest share. Since adiabatic reaction operation requires high methane to oxygen ratios, only a small amount of oxygen is consumed. Reducing biogas production cost, e.g., by improving its generation via AD or its treatment step, can yield more significant cost savings to the process.

The obtained ethylene to ethane ratio in the reactor is 1.5, so a significant production of ethane is achieved. However, revenue from ethane sales has a very limited contribution to the total cash flow due to its low sales price. Recycling ethane back to the Bio-OCM reactor could improve ethylene production and lead to additional revenue, but it is likely to reduce overall process selectivity as both $C_2$ products are readily oxidized into $CO_2$. A potentially good alternative is the addition of Ethane Dehydrogenation (EDH) thermal or catalytic reactor that can selectively convert ethane into ethylene and hydrogen. This is not evaluated further to avoid over-extending this study, but should be considered in future analyses.

The lights stream is found to be the most sensitive component of the total cash flow. Since methane conversion in the reaction is low, a significant amount of methane is exported in this side-product stream. In fact, the entire process concept can also be seen as a biogas upgrade plant producing ethylene as a side product. Therefore, the total cash flow is highly sensitive to the price adopted for the lights stream. For the nominal and best-case scenarios, a positive cash flow is obtained even without bio-ethylene sale revenues. For the worst-case scenario, however, the cash flow is negative, thus increasing the production cost of bio-ethylene.

Recycling the lights stream back to the OCM reactor is undesired because it also contains $H_2$, CO, and $N_2$. Treating this stream to recycle only the methane portion is possible, but energy-intensive. Another alternative is the use of a methanation (Sabatier) reactor, within which CO and $H_2$ from the lights stream and $CO_2$ from the absorption separation section are converted into more methane. Finally, this stream could also be reformed with carbon dioxide, e.g., methane dry reforming, for the side production of synthesis gas. These alternatives could be considered in future techno-economic evaluations as alternative pathways for valorizing the lights off-gas stream.

### 4.2. Utility Cost

Table 4 provides the total utility cost rates for each utility category and for each section of the Bio-OCM process. Electricity consumption, which incurs mostly from gas compression, represents the major source of utility cost. Refrigeration cost is also indirectly comprised mostly of electricity to run refrigeration compressors. Altogether, these two categories amount to nearly 62% of the total utility cost rate. Therefore, the adopted RSV distillation configuration offers great economic benefit to the process by providing cost savings in the most critical utility category.

Medium Pressure Steam (MPS) represents the second most expensive utility category. MPS is used solely in the amine regeneration step, which makes this the single most expensive step in the entire process representing 52% of the total utility cost rate. This is due to the large amount of $CO_2$ that enters the process with the biogas. This study considered the use of a benchmark 30 wt% MEA solution, which reacts extremely fast with

$CO_2$, but requires significant regeneration energy. In this sense, switching to another amine solution could potentially lead to significant cost savings.

Regarding the different process sections, the $CO_2$ removal represents the highest share of utility cost rate. Besides MPS for amine regeneration, this section also consumes a significant amount of electricity to compress the gas from 1.3 bar to 3.13 bar as well as cooling water to run the condenser of the regeneration column, the other amine coolers, as well as the inter-stage cooling of the gas compressors. The distillation section consumes mostly electricity to run the process and refrigeration compressors, whereas the reaction section is able to export HPS to generate additional revenue.

**Table 4.** Yearly cost rates for the different utility in each process section. Positive values mean consumption, while negative values mean export.

| Process Section Utility Category | Reaction kUSD Year$^{-1}$ | CO$_2$ Removal kUSD Year$^{-1}$ | Distillation kUSD Year$^{-1}$ | TOTAL kUSD Year$^{-1}$ |
|---|---|---|---|---|
| Electricity | - | 945 | 917 | 1862 |
| LPS | - | - | 20 | 20 |
| MPS | - | 1713 | - | 1713 |
| HPS | −606 | - | - | −606 |
| Refrigeration | - | - | 154 | 154 |
| Cooling Water | - | 108 | 6 | 114 |
| TOTAL | −606 | 2766 | 1098 | 3258 |

### 4.3. Equipment Cost

Table 5 details the cost of the different equipment types installed in each process section. Compressors represent the most expensive equipment category closely followed by refrigeration equipment, which also mostly consists of refrigeration compressors. Altogether, these two categories represent 56% of the total capital investment in equipment. The third largest category is heat exchangers, followed by columns, drums, and then by minor contributions of furnaces, reactors, and pumps.

Among the different process sections, distillation is presents the highest capital investment, accounting for 61% of the installed equipment cost. This is mainly because of process gas and refrigeration compressors. It is followed by the $CO_2$ removal section, and finally by the reaction section.

**Table 5.** Total cost of different equipment installed in each process section.

| Process Section Equipment Category | Reaction kUSD | CO$_2$ Removal kUSD | Distillation kUSD | TOTAL kUSD |
|---|---|---|---|---|
| Compressors | - | 1075 | 5518 | 6593 |
| Heat Exchangers | 908 | 1340 | 976 | 3225 |
| Furnaces | 810 | - | - | 810 |
| Reactors | 538 | - | - | 538 |
| Columns | - | 1578 | 315 | 1895 |
| Drums | - | 1112 | 485 | 1596 |
| Pumps | - | 167 | 67.3 | 234 |
| Refrigeration | - | - | 4228 | 4228 |
| TOTAL | 2256 | 5274 | 11,590 | 19,119 |

The OCM reactors are assumed to be carbon steel vessels with a refractory lining and packed with the catalyst as described in recent patents [42]. While the reactor by itself is not a mechanically complex and expensive equipment to manufacture, there would be additional charges to cover the cost of technological development, e.g., in the form of royalties. This uncertainty is assumed to be accounted for in the Monte Carlo simulation, wherein the equipment cost is varied between −30% and +50% as discussed in Section 2.4.5.

*4.4. Bio-Ethylene Production Cost*

The total bio-ethylene product output is $4632\,t_{C_2H_4}\,year^{-1}$. Based on the cost estimations obtained in Sections 4.1–4.3, the production cost per mass of bio-ethylene is calculated using Equation (13). The estimated values and ranges of each cost components are used to compute a nominal as well as a worst and a best-case scenarios. These are reported in Table 6.

**Table 6.** Bio-ethylene production cost for different cost scenarios. Positive values mean cost, while negative values meaan revenue.

| Scenarios Cost Components | Worst | Nominal | Best |
|---|---|---|---|
| | \multicolumn{3}{c}{USD $kg_{C_2H_4}^{-1}$} | |
| Utility | 0.91 | 0.70 | 0.49 |
| Annualized Equipment | 0.92 | 0.61 | 0.43 |
| Educts & Products | 0.42 | −0.80 | −2.03 |
| TOTAL | 2.25 | 0.51 | −1.11 |

Table 6 clarifies that a wide range of results are possible and that the main source of uncertainty relates to the cost of educts and and price of side-products. More specifically, the most sensitive parameter is the cost assigned to the lights stream, which depends on the cost of natural gas in Brazil. In the best-case scenario, wherein the light stream is sold at $5.0\,USD\,GJ_{HHV}^{-1}$, the revenue from side products is high enough to completely cover all expenses, resulting in a negative bio-ethylene production cost. In this scenario, bio-ethylene revenue would generate a bonus. In the worst-case scenario, however, the lights stream is sold at $2.0\,USD\,GJ_{HHV}^{-1}$ and the resulting production cost for bio-ethylene is above typical market values for fossil ethylene. The nominal case results in a production cost below market value.

Table 6 also highlights a well-distributed share between utility and equipment cost. Scaling-up the system would typically lead to a further beneficial dilution of the equipment cost due to economies of scale, but this obviously difficult since biogas production facilities tend to be rather distributed. Therefore, large-scale bio-ethanol plants generating significant amounts of biogas through vinasse AD are the best candidates for this type of project. The ethylene production capacity is, however, still tiny compared to typical fossil-based naphtha or ethane steam cracking plants.

*4.5. Monte Carlo Simulation*

The Monte Carlo simulation with 10,000 samples resulted in a production cost of $0.53 \pm 0.73\,USD\,kg_{C_2H_4}^{-1}$. The average bio-ethylene production cost is, therefore below typical market values for fossil ethylene. However, the standard deviation clarifies again how wide the range of outcomes is.

The cumulative distribution for the bio-ethylene production cost resulting from the Monte Carlo simulation is given in Figure 15. The line shows in how many of the Monte Carlo samples (percent-wise) the calculated bio-ethylene production is less or equal to the values given in the x axis. It can be understood as the confidence that the production cost is less or equal to the given value. For instance, a bio-ethylene production costs lower than fossil ethylene's lowest market value of $0.70\,USD\,kg_{C_2H_4}^{-1}$ is achievable with a 55.2% confidence, whereas a production cost lower than ethylene's highest market value of $1.50\,USD\,kg_{C_2H_4}^{-1}$ can be achieved with 87.0% confidence. For confidence levels higher than 95%, the bio-ethylene production cost is way above the market value. Overall, it becomes clear the bio-ethylene cannot provide an economic alternative to fossil ethylene with a high degree of confidence.

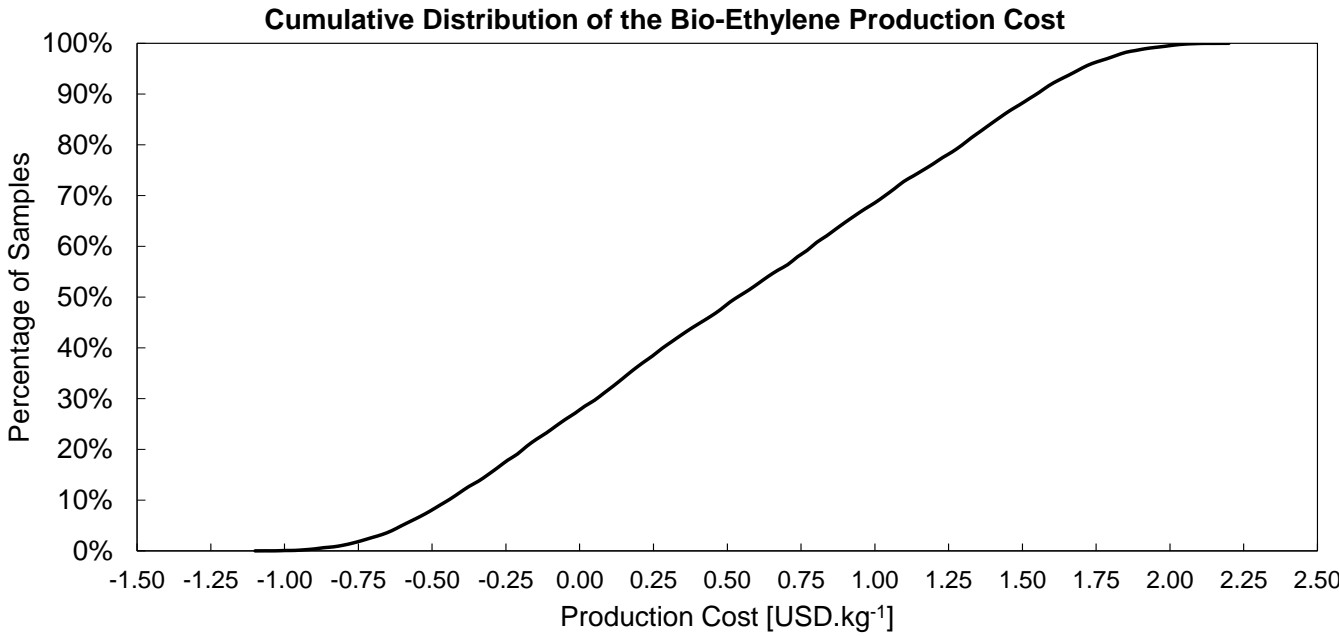

**Figure 15.** Cumulative distribution for the bio-ethylene production cost resulting from the Monte Carlo simulation.

### 4.6. Discussion

Brazil produces ethylene mainly from steam cracking of naphtha. Even though the country has recently become a net oil exporter [53], it imports light oil for petrochemical production ($\approx$113 million Barrels of Oil Equivalent (BOE) in 2012) given that internal production consists mostly ($\approx$70%) of heavy oil that cannot be processed effectively in the domestic refinery park [54]. Hence, production cost of fossil ethylene in the country is speculated to be on the higher end of the spectrum.

Brazil also hosts a 200 kt year$^{-1}$ bio-ethylene plant, which is obtained by catalytic dehydrogenation of bio-ethanol [55]. It is estimated that the cost of bio-ethylene production from first generation bio-ethanol in Brazil is in the range of 1.13 USD kg$^{-1}_{C_2H_4}$ to 1.17 USD kg$^{-1}_{C_2H_4}$ (considering the average December of 2019 exchange rate of 1.11 USD EUR$^{-1}$) [55], which is in accordance with the cost range estimated in this work. The manufacturer, i.e., Braskem, uses it to make polyethylene sold under a seal denominated "I am green". The products holding this seal must have their bio-based carbon content determined via radiocarbon analysis (ASTM D6866) [56]. It is unclear, yet highly possible, that customers would accept paying a higher price for a certified bio-based polyethylene containing the seal. This type of certification and market strategy could also enable bio-ethylene produced via Bio-OCM to reach higher market values.

The depletion of resources and introduction of restrictive measures such as carbon taxation is also likely to gradually raise the price for fossil commodities, including ethylene and natural gas, in short-to-medium term. Fully replacing fossil-based ethylene by biogas-based ethylene is unthinkable due to the limited production volumes achievable. Bio-OCM can, however, similarly to catalytic bioethanol dehydration, act as a bridge technology. It immediately allows for the use of a renewable feedstock to produce traditional platform chemicals with established processing and transportation infrastructure until the production of new bio-based platform chemicals is fully developed and scaled-up.

This study also highlighted the importance of the lights stream as a side-product for the overall economic viability of the process. Several OCM researchers are trying to improve yield of C$_2$ products via new catalysts and novel reactor concepts, e.g., chemical looping [57]. However, industrial implementation of such technologies only seem feasible in the medium-to-long term future, while developing multi-product systems that efficiently integrate OCM can offer significant potential for industrial deployment today. Future studies

should, thus focus in making the most out of the off-gas stream, e.g., via methanation and methane dry reforming. A combination of these process routes would be a solution to increase flexibility in the production of bio-ethylene, bio-syngas, and bio-energy based on fluctuating market conditions.

The vinasse effluent is currently employed within the agricultural sector for soil and water amendment in a technique called fertigation. However, if its high organic load is not properly treated prior to this, significant GHG (particularly methane) emissions incur. Providing alternative pathways for monetizing biogas produced through vinasse AD adds incentive for its treatment and for emission reductions. Based on the yields obtained in this study and the biogas production potential of the state of São Paulo, Brazil [23], it has been estimated that ethylene produced through Bio-OCM could replace as much as 900 kt of fossil ethylene yearly in this state alone. This could potentially also lead to significant environmental benefits, which are best addressed through more holistic studies, i.e., Life Cycle Assessment.

## 5. Conclusions and Outlook

This contribution designs a process and investigates the economic potential of bio-ethylene production via Oxidative Coupling of Methane (OCM) contained in biogas derived from the Anaerobic Digestion (AD) of vinasse effluent from bioethanol refineries in Brazil. The optimized process structure consists in a reaction section employing Packed Bed Reactors (PBRs) in adiabatic regime, a $CO_2$ removal section employing amine-absorption, and a distillation section employing a Recycle Split Vapor (RSV) configuration. A $C_2$ reaction product yield of 16.12% is achieved in the simulations by optimizing process conditions. However, technical challenges regarding long-term catalyst and operation stability under the high temperature and $CO_2$ dilution involved still need to be addressed, and the potential use of low-temperature catalysts should be considered in future studies aiming at industrial implementation. The proposed hybrid membrane-absorption configuration for the $CO_2$ removal section is found to be uneconomical. The proposed RSV distillation configuration, on the other hand, provides a 24% lower total annualized cost than the traditional distillation configuration and, thus, has a great potential to be employed industrially.

The bio-ethylene production cost is estimated under a wide range of possible costs for educts, products, utilities, and equipment by means of a Monte Carlo simulation. The resulting average production cost of bio-ethylene is $0.53 \pm 0.73$ USD $kg_{C_2H_4}^{-1}$ and is lower than typical ethylene market values. The bio-ethylene production cost is very sensitive towards the price assigned to the lights stream as a side-product, which in this study is based on the natural gas price. Under the assumed range of price scenarios, a bio-ethylene production cost under 0.70 USD $kg_{C_2H_4}^{-1}$ is only achieved with a 55.2% probability and in those cases where natural gas, ethylene, and/or electricity prices are high. Further research should focus on reducing the uncertainty on the price assigned to the lights stream as well as investigating different process integration options to exploit this stream.

**Author Contributions:** Conceptualization, A.T.P., H.R.G., G.L. and J.A.D.R.; methodology, A.T.P., E.E. and J.-U.R.; software, A.T.P. and E.E.; validation, A.T.P.; formal analysis, A.T.P., H.R.G., G.L., A.P.O. and J.-U.R.; investigation, A.T.P., G.L. and A.P.O.; resources, A.T.P. and G.L.; data curation, A.T.P. and G.L.; writing—original draft preparation, A.T.P. and G.L.; writing—review and editing, all; visualization, A.T.P.; supervision, J.-U.R., J.A.D.R., A.O. and J.G.; project administration, J.-U.R., J.A.D.R., A.O. and J.G.; funding acquisition, all. All authors have read and agreed to the published version of the manuscript.

**Funding:** This research and the BIOCM project were funded by the ERANet-LAC 2nd joint call for proposals, with grants from the German Federal Ministry of Education and Research (BMBF 01DN17023), The Brazilian National Council for Scientific and Technological Development (CNPq 443181/2016-0), Colombian Minciencias (contrato 483-2016) and The State Education Development Agency of Latvia (ES RTD/2017/23). Alberto Penteado gratefully acknowledges funding from CAPES/Brazil (11946/13-0) and Giovanna Lovato gratefully acknowledges the funding from

FAPESP/Brazil (2014/07692-8 and 2018/17715-6). We acknowledge support by the German Research Foundation and the Open Access Publication Fund of TU Berlin.

**Institutional Review Board Statement:** Not applicable.

**Informed Consent Statement:** Not applicable.

**Data Availability Statement:** The data presented in this study are openly available in DepositOnce Repository for Research Data and Publications from TUB (10.14279/depositonce-12115 and 10.14279/depositonce-12193).

**Conflicts of Interest:** The authors declare no conflict of interest.

## Abbreviations

The following abbreviations are used in this manuscript:

| | |
|---|---|
| AD | Anaerobic Digestion |
| APEA | Aspen Process Economic Analyzer |
| Bio-OCM | Biogas-based Oxidative Coupling of Methane |
| BOE | Barrels of Oil Equivalent |
| CHP | Combined Heat and Power |
| COD | Chemical Oxygen Demand |
| DAEs | Differential Algebraic Equation System |
| EDH | Ethane Dehydrogenation |
| EoS | Equation of State |
| GHG | Greenhouse Gases |
| GSM | Gas Separation Membranes |
| HPS | High Pressure Steam |
| LPS | Light Pressure Steam |
| MPS | Medium Pressure Steam |
| eNRTL | Electrolyte Non-Random Two-Liquid |
| OCM | Oxidative Coupling of Methane |
| MEA | Monoethanolamine (IUPAC: 2-aminoethan-1-ol) |
| PBR | Packed Bed Reactor |
| PFR | Plug-Flow Reactor |
| PPG | Plastic Pyrolysis Gas |
| PR | Peng-Robinson |
| PSA | Pressure-Swing Adsorption |
| RSV | Recycle Split Vapor |
| TUB | Technische Universität Berlin |

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
