# Peer review of "Economic Potential of Bio-Ethylene Production via Oxidative Coupling of Methane in Biogas from Anaerobic Digestion of Industrial Effluents"

_processes, doi:10.3390/pr9091613_

Round 1
Reviewer 1 Report
This paper designed a chemical process and studied the potential of oxidative coupling of methane from biogas (anaerobic digestion of vinasse from bioethanol refineries in Brazil) to produce bio-ethylene. The analysis is detailed and through. The writing is easy to read and follow. I recommend this paper to be accepted.
- On Page 6 Line 176, “poliymide” should be polyimide.
Author Response
Dear reviewer,
I would like to thank you in the name of all authors for taking your time to review and provide valuable comments to our manuscript. Please find answers to your points below:
- On Page 6 Line 176: “poliymide” changed to “polyimide”.
Yours most sincerely,
Alberto T. Penteado,
Reviewer 2 Report
The paper “Economic Potential of Bio-Ethylene Production via Oxidative Coupling of Methane in Biogas from Anaerobic Digestion of Industrial Effluents” is interesting and organized but to improve the quality the following recommendations can be incorporated:
- In my opinion the chemical reactions from (1) to (10) can be remove.
- At line 146 and in the figure 1 title please use only one measure unit for temperature.
- It will be interesting to update the table 2 by inserting the characteristic of conventional or unconventional fuels (please see https://doi.org/10.37358/MP.19.4.5259).
- Please remove the gridlines from figures 5, 7, 8, 13, 14.
- Please check and rewrite the references according to journal instructions.
Author Response
Dear reviewer,
I would like to thank you in the name of all authors for taking your time to review and provide valuable comments to our manuscript. Please find answers to your points below:
- We have considered this, however, we think that showing the entire reaction network (including side-reactions) is essential for the readers to fully understand the complexity of the process and why the downstream separation units are required to purify the main product bio-ethylene.
- Figure 1 does not have any temperature units in the title. I think you are referring to Figure 2, which showed the reaction temperature in both Celsius and Kelvin. We have modified the caption pf Figure 2, which now only shows temperature in Celsius (same as in the plot).
- We have added the characteristics of Biogas and of Plastic Pyrolisis Gas obtained from catalytic pyrolysis of high-density polyethylene to the table and referred to them in the text. The excellent source article has been cited appropriately.
- We have considered this, but we believe the grid-lines are helpful to read the plot
- We have checked the references and citation style. We have used the Journal's template and followed the instructions by using numbered references placed in square brackets.
Yours most sincerely,
Alberto T. Penteado
Reviewer 3 Report
This is a good piece of paper with a good title. The overall presentation, content, scientific soundness of the paper is good.
There could have a section on further explanation of anaerobic digestion specially the four phases of anaerobic digestion - Hydrolysis, Acidogenesis, acetogenesis and methanogenesis.
Author Response
Dear reviewer,
I would like to thank you in the name of all authors for taking your time to review and provide valuable comments to our manuscript. We have checked and revised English language misspellings and modified the introduction by adding a paragraph explaining the four phases of anaerobic digestion.
Yours most sincerely,
Alberto T. Penteado